# Unveiling the transition from niche to dispersal assembly in ecology

Lynette H. L. Loke[1✉] & Ryan A. Chisholm[2]

A central goal in ecology is to understand what maintains species diversity in local communities. Classic ecological theory[1,2] posits that niches dictate the maximum number of species that can coexist in a community and that the richness of observed species will be below this maximum only where immigration is very low. A new alternative theory[3,4] is that niches, instead, dictate the minimum number of coexisting species and that the richness of observed species will usually be well above this because of ongoing immigration. We conducted an experimental test to discriminate between these two unified theories using a manipulative field experiment with tropical intertidal communities. We found, consistent with the new theory, that the relationship of species richness to immigration rate stabilized at a low value at low immigration rates and did not saturate at high immigration rates. Our results suggest that tropical intertidal communities have low niche diversity and are typically in a dispersal-assembled regime where immigration is high enough to overfill the niches. Observational data from other studies[3,5] suggest that these conclusions may generalize to other ecological systems. Our new experimental approach can be adapted for other systems and be used as a 'niche detector' and a tool for assessing when communities are niche versus dispersal assembled.

A central unresolved question in ecology concerns the factors that drive and maintain species diversity in local communities. Under niche-assembly theory, species richness is explained in terms of stable coexistence attributable to properties of the local environment: species differ in their niches and these differences collectively stabilize community dynamics. Under dispersal-assembly theory, including MacArthur and Wilson's theory of island biogeography[1,2], the neutral theory of biodiversity[6] and mass effects[7], local diversity arises from an immigration–extinction balance and is ultimately dependent on regional processes.

Which of these two distinct perspectives is more correct is likely to depend on the context[8–13]. One critical mediating factor is the immigration rate to a community. Under pure niche assembly, there should be no relationship between species richness and the immigration rate (Fig. 1a). Under pure dispersal assembly, the relationship should be monotonically increasing (Fig. 1b). A classic unification of the two theories postulates that dispersal assembly applies only in cases where immigration is too low to fill all the niches (Fig. 1c). Indeed, MacArthur[14] and Wilson[15] speculated that the immigration–extinction balance and dispersal assembly central to their theory of island biogeography[1,2] applied only below some threshold immigration rate, above which species richness would be determined mainly by habitat diversity, that is, niche assembly[14] or by the diversity of the source species pool[15]. Under this hypothesis, the relationship between species richness and the immigration rate is predicted to increase for low immigration rates and then saturate at high immigration rates[15–17] (Fig. 1c). This saturation point reflects the maximum possible number of species or the niche

diversity of the island or patch—a ceiling that cannot be exceeded even with greater levels of immigration[14].

A recently formulated new unified hypothesis[3,4], however, predicts just the opposite: that niche assembly applies where immigration is low and dispersal assembly applies where immigration is high (Fig. 1d). Related ideas go back to Diamond's community assembly rules[18], the core–satellite hypothesis[19] and source–sink dynamics[20]. Under this hypothesis, niches stabilize diversity at some fixed level and provide a floor on species richness where immigration is low, but, as immigration increases, communities undergo a transition from a niche-assembled regime to a dispersal-assembled regime (Fig. 1d). The relationship of species richness to immigration rate is predicted to be nearly flat for low immigration rates and increasing for high immigration rates (Fig. 1d): that is, species richness asymptotes at low immigration rates rather than high immigration rates as in the classic hypothesis (Fig. 1c). The transition point occurs where immigration becomes sufficient to overwhelm the niche constraints, which leads to a dynamic immigration–extinction equilibrium with more species than niches.

Which of these two contrasting unified hypotheses is correct has fundamental consequences for our understanding of the forces that structure ecological communities. The two hypotheses make very different predictions about the shape of the immigration–species richness curve, but there has not yet, to the best of our knowledge, been any experimental test to discriminate between them. A key challenge in experimentally testing the transition between dispersal- and niche-assembled regimes is the lack of foreknowledge as to the range over which the immigration rate should be varied to reveal

[1]School of Natural Sciences, Faculty of Science and Engineering, Macquarie University, North Ryde, New South Wales, Australia. [2]Department of Biological Sciences, National University of Singapore, Singapore, Singapore. ✉e-mail: lynetteloke@gmail.com

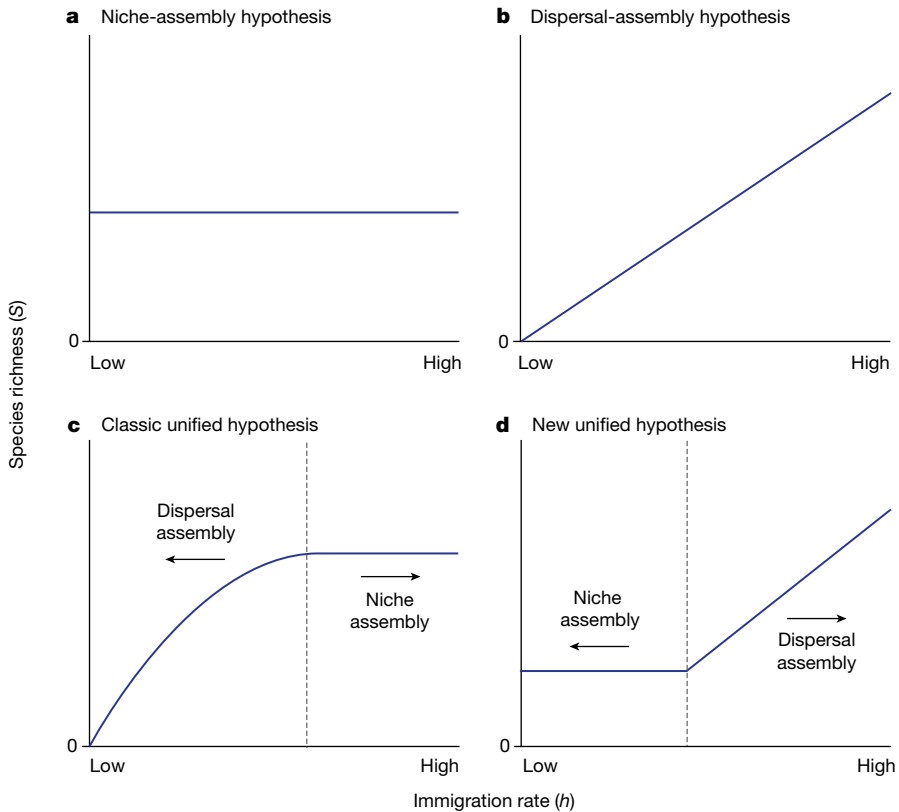

**Fig. 1 | Theoretical predictions for the relationship of species richness to immigration rate in ecological communities. a**, The niche-assembly hypothesis predicts constant species richness (*S*) with immigration rate (*h*). **b**, The dispersal-assembly hypothesis predicts a monotonically increasing relationship. **c**, A classic unifying hypothesis proposes that dispersal assembly applies under low immigration and niche assembly at high immigration. **d**, A new unifying hypothesis predicts the opposite. The dashed lines in panels **c** and **d** represent the transition points between the two assembly regimes.

both regimes (Fig. 1). In deterministic theoretical models, the niche-assembled regime can be uncovered by simply setting immigration to zero, but, in stochastic models or real experimental settings, this strategy fails because even in the niche-assembled regime, occasional immigrants are needed to offset stochastic extinctions. One feasible experimental approach is to vary the immigration rate across treatments with the rate approaching but not equal to zero in the low-immigration treatments. This is analogous to the mathematical approach of taking the limit as immigration tends to zero to avoid a singularity at zero. Ecologists acknowledge that immigration can in general influence community species richness[2,7,11], but the key to our experimental approach is that we systematically vary immigration over a wide range in this way to reveal the functional form of the species richness versus immigration relationship and thus explain underlying mechanisms. Here, we use this approach to conduct a crucial experimental test of the predictions of the classic and new hypotheses to assess the conditions under which ecological communities are primarily niche versus dispersal assembled (Fig. 1).

## Experimental design

We performed a manipulative field experiment on intertidal communities[21–23]. We focused on intertidal seawalls—a well-studied tractable model system[24–26] (Methods). We experimentally manipulated the level of immigration across experimental habitat patches using custom-built set-ups (Fig. 2). Each set-up was a treatment replicate with two components: (1) a topographically complex concrete habitat tile that provided a standardized level of niche diversity across all replicates (Fig. 2a and Methods); and (2) a stainless-steel cage, with clear square polycarbonate sheet panels (slotted into the cage), that fitted over each

tile to create a seal to ensure that the immigration of organisms to and from each unit was only via fixed-size holes (40 mm diameter) cut into the clear panels (Fig. 2b and Methods). Each set-up was submerged at high tide (95.2% of the time) and exposed at low tide (4.8% of the time; see Methods). We created a total of 12 treatments by varying the number of holes (*h*) in the clear polycarbonate panels (Fig. 2c).

We installed a total of 60 experimental units (12 *h*-treatments × 5 replicates) in randomized order along the base of intertidal seawalls (approximately 0.5 m above chart datum) across a 400 m stretch at our study site in Singapore (Methods). To maintain the set-ups over the duration of the experiment, polycarbonate panels were removed and replaced with clean new ones every two to four weeks during low tide. We censused the number of species, species identities and abundances of the macrofaunal intertidal communities within each set-up every month for a year (Methods). Although we focused on macrofaunal intertidal species for practical reasons, we expect that qualitatively similar results would be obtained if we included algal and microbial communities. In total, we recorded 10,156 individuals of 64 different species: major taxa included gastropods, bivalves, polychaetes, tunicates and crustaceans (see full list in Supplementary Table 1). Species richness stabilized in all but the highest immigration treatments after six months (Extended Data Fig. 1), consistent with findings from previous studies carried out at the same study site and at nearby locations[25–27]. We thus used data from months 7 to 12 in our analyses (Extended Data Fig. 1). We fitted mechanistic mathematical models corresponding to each of the two hypotheses to the data by generalized nonlinear least squares (Methods). The classic model has two fitted parameters (the ratio of the immigration and extinction rates and metacommunity richness) and the new model has three (immigration rate parameter, niche diversity and the fundamental biodiversity

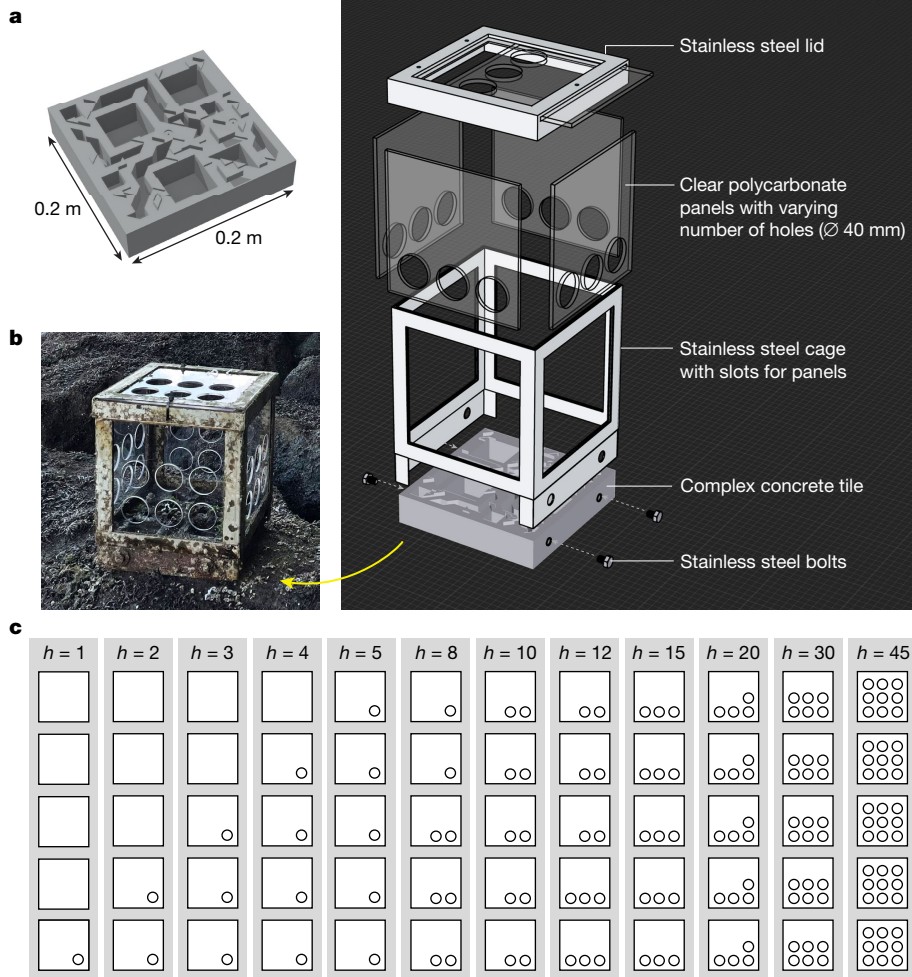

**Fig. 2 | Experimental set-up for manipulating the immigration rate to an intertidal community. a,b,** Each set-up comprised two components: a habitat tile, used to standardize habitat surface area and structural complexity across replicates (**a**); and a custom-built cage that fitted securely over the habitat tile to create a seal, with slots that allowed five clear polycarbonate sheet panels to be inserted and removed for regular maintenance (**b**). **c,** The level of immigration was controlled by varying the total number of holes $h$ on the clear polycarbonate panels. We had 12 $h$-treatments (grey boxes; $h$ = 1, 2, 3, 4, 5, 8, 10, 12, 15, 20, 30, 45) and five replicates ($n$ = 5) of each treatment set-up. The five white squares within each grey box show the number of holes and their arrangements for each of the five panels used to create the corresponding treatment. Panels in the first row were the top panels.

number; see Methods). We assumed that immigration was linearly related to the number of holes ($h$), a parsimonious assumption that, if relaxed, only strengthens our results (Methods). We performed 1,000 bootstraps stratified by hole treatments to obtain 95% confidence intervals on the parameter estimates and fitted values (Methods).

## Evidence of the transition

The relationship of species richness to immigration rate across experimental treatments was consistent with the new hypothesis, with a nearly flat phase at low immigration rates and an increasing phase at high immigration rates (Fig. 3; Akaike information criterion, AIC = 974.0). The fit of the classic model to the data was substantially poorer (Fig. 3, AIC = 1066.3; see also Extended Data Fig. 2, for a variant of the classic model). The new model explained 83% of the variation in the data. This provides experimental evidence of the transition from niche to dispersal assembly in ecology. The estimated niche diversity was $n^* = 4.4$ with a 95% confidence interval of [4.2, 5.2], indicating that approximately four or five species can stably coexist in the seawall tile communities in the absence of substantial immigration, with species richness close to this minimum value being realized in the lowest immigration treatments ($h$ = 1, 2, 3, 4 and 5). The transition to the dispersal-assembly regime occurred at intermediate immigration rates ($h \approx 10$) and roughly three times the minimum number of species were present in the highest immigration treatment ($h$ = 45). The effect of increasing species richness was partly mediated by increasing total abundance with immigration among the high-immigration treatments and partly by increasing species richness given total abundance (Fig. 4).

## Discussion

Our experimental results are consistent with the new hypothesis, that is, that communities are niche assembled under low immigration and are otherwise primarily dispersal assembled[3,4]. This is consistent with previous observational evidence of a biphasic species–area relationship in island archipelagos, where area is interpreted as a proxy for immigration[3]. However, the island data, being observational, are subject to alternative interpretations[5] because confounding variables, such as niche diversity, can covary with area[7]. Our experimental approach avoids this by standardizing area across all treatments and thus provides a more direct test of theoretical principles about the effects of immigration on community assembly. We note that species richness was still increasing slowly and did not fully equilibrate in our highest immigration treatments ($h$ = 30, 45) after six months (Extended

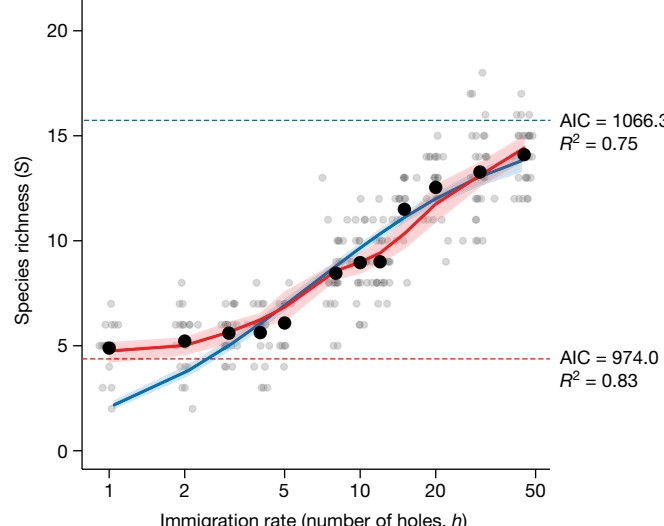

**Fig. 3 | Evidence of the transition from niche to dispersal assembly.**
Each small grey point indicates a single experimental unit (seawall tile; 12 $h$ -treatments, $n$ = 5) in a given month ($x$ coordinates are jittered to improve visibility); each large black point shows a treatment mean. Fitted models are shown by the solid curves (mean estimates) with shaded 95% confidence intervals from 1,000 bootstraps; the new model (equation (6)) is shown in red whereas the classic model is in blue (equation (3); see Methods). Consistent with the new model, the relationship of species richness to immigration in experimental intertidal seawall communities is biphasic and asymptotes to a low value at low immigration rates, indicating that, for low immigration rates, communities are primarily niche assembled and comprise approximately four or five species (dashed red horizontal line) and that, for high immigration rates, communities are primarily dispersal assembled. Under the new model, hypothetical species richness due to niche assembly alone is constant and close to richness in the low-immigration treatments (dashed red horizontal line). The classic model was substantially poorer than the new model (judged by Akaike information criterion, AIC). The dashed blue horizontal line indicates the expected species richness at high levels of immigration under the classic model, as estimated by the fitted model.

Data Fig. 1): allowing it to equilibrate would only strengthen our conclusions by accentuating the biphasic relationship of species richness to immigration rate (species richness would be higher towards the right of Fig. 3).

Our new experimental approach can be adapted to other systems such as annual plant communities[28] and microbial communities[29] in laboratory conditions. In addition to providing additional tests of

our two central hypotheses (Fig. 1c,d), the experimental approach can be used to assess when communities are niche versus dispersal assembled—thereby shedding light on a question that has perplexed ecologists for decades. The experiments also act as a 'niche detector', that is, a tool for assessing how many species can coexist without substantial immigration—another long-standing ecological conundrum.

Our experimental data, taken together with past observational and theoretical studies, suggest that niche diversity in ecological systems is typically low. The estimated niche diversity of our intertidal seawall model system was low ($n$* = 4.4) and we predict that similar values would be obtained in experiments on the natural analogues of our system, tropical rocky shores. Low niche diversity for a variety of ecological communities is also implied by island species–area relationships, which typically exhibit a similar shape to our species–immigration relationship (Fig. 3), with a flat low-richness phase at low island areas attributable to niche assembly under low immigration[3]. This flat phase has long been known as the 'small-island effect'[30,31], but lacked a satisfactory dynamical explanation until recently[3]. Theoretical models also suggest that the number of species that can coexist via local niche mechanisms is generally low. Although, in some theoretical models, a large number of species can stably coexist via temporal niches arising from relative nonlinearity[32,33] and storage effects[34], such coexistence is fragile and not robust to stochasticity and perturbations, which raises doubt about its relevance in nature[35].

We predict that applications of our experimental approach to other systems will lend further support to the hypothesis that niche diversity is generally much lower than the typical total species richness of natural habitats, which implies that most species are not stably coexisting but, instead, are transiently co-occurring and are reliant on continued immigration[8] (that is, mass effects[7]). A corollary is that most natural communities are in the dispersal-assembly regime. In rocky intertidal communities characteristic of our study site, species richness is invariably much higher than observed in our niche-assembly regime[36,37]. We conjecture that these conclusions also apply to highly diverse systems such as tropical rainforest tree communities, where hundreds of species may be present in a single hectare[38]. A prominent niche-assembly explanation for high tree species richness is that predators and pathogens are largely host-specific, which leads to a high diversity of enemy-escape niches[39]. By contrast, the dispersal-assembly explanation attributes high local tree species richness to immigration, as well as to potentially larger-scale niche processes (for example, source–sink dynamics[20], with species being adapted to distinct habitats across the landscape) and, ultimately, to diversification processes that play out over geological timescales and continental spatial scales. Although our experimental approach would be impractical in tropical forests because of long tree generation times, the question of what determines high tree diversity

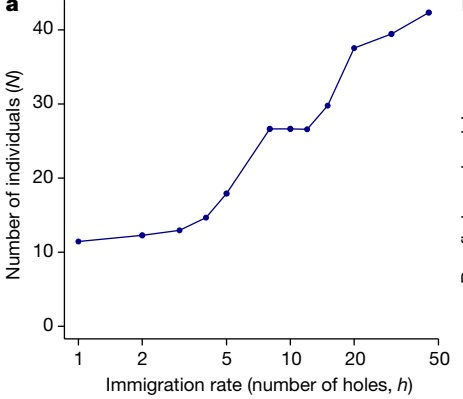

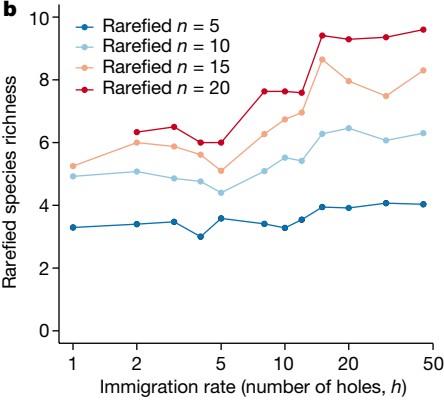

**Fig. 4 | As immigration is increased, both increasing number of individuals and increasing rarefied species richness contribute to increasing species richness (Fig. 3). a,** Average number of individuals ($N$) for each value of immigration ($h$). **b,** Average rarefied species richness for each value of immigration ($h$), using various rarefaction sample sizes ($n$).

could be indirectly informed by evidence from other, more experimentally tractable, high-diversity communities, such as plankton communities[33,40]. If plankton communities collapse to a small number of species under low immigration, similar to our seawall communities (Fig. 3), this would cast doubt on the general notion that an observation of high species diversity implies high niche diversity.

Given the evidence for two distinct assembly regimes, we encourage future experimental work—whether on intertidal systems or other communities—to carefully distinguish which regime prevails in any given context and to study each regime separately. In the niche-assembly regime, although we observed some consistency in the set of coexisting species (for example, the gastropod *Drupella margariticola* and the tunicate *Didemnum psammatodes*), our experimental set-up was not designed to unravel the precise niche-assembly rules. Figuring out the exact niche mechanisms operating in this regime is a priority for future research. Experiments that directly manipulate the species identities of immigrants can potentially address this (as done in microbial communities[29]). An alternative would be to spread the experimental units over a much larger geographical area to capture a larger species pool. For our study system, our estimate of the transition point between the two regimes ($h \approx 10$) can inform the design of future studies focused on one regime or the other. Our results can also inform sample size determination for future studies focusing on more-detailed biodiversity patterns including co-occurrence matrices and species abundance distributions. One assumption of our method is consistency of abiotic conditions across experimental immigration treatments. We confirmed this for light and temperature (Methods), but it remains conceivable that other variables (for example, pH) are important and we encourage future similar experiments to measure as wide a range of abiotic variables as possible. More broadly, to understand the dispersal-assembly regime, ecologists must design more cross-scale studies that link local and regional dynamics: it is increasingly clear that the factors driving the diversity of a small local community cannot be unravelled by studying the system in isolation, that is, without accounting for immigration[9,11]. A holistic assessment of the dispersal-assembly regime should also account for niche processes that may not on their own permit stable coexistence (as in the niche-assembly regime) but nevertheless slow competitive exclusion and thus act synergistically with immigration to increase species richness[41,42]. Such synergies between immigration and niche processes may turn out to be more important for diversity than traditional stable coexistence.

Our results have practical implications for the use of concrete tiles to augment diversity on man-made coastal defences in many parts of the world—a strategy known as eco-engineering or nature-based solutions[43,44]. The predominant role of dispersal assembly in our system indicates that such efforts may be ineffective unless attention is also paid to habitat preservation and connectivity in the broader landscape to ensure a large, diverse, continued supply of immigrants. Regardless of their topographic complexity, concrete tiles placed in a depauperate intertidal landscape will harbour few species and this needs to be considered more explicitly, especially in real-world applications. More broadly, the importance of dispersal assembly and the apparent low number of stabilized niches cautions against conservation approaches that rely too strongly on niche mechanisms for preserving species diversity[45]. Efforts to protect biodiversity must account for large-scale landscape connectivity to be successful—although we encourage further experimental tests of the new hypothesis in other systems and locations to establish the generality of this conclusion.

The emerging picture, not only from our experimental study but also from observations on island archipelagos and from theoretical work, is that dispersal assembly is the prevailing regime for most ecological communities and that the number of stabilized niches is low. This may explain the relative success of niche theory in low-diversity experimental systems[28,46,47] but its failure to provide a predictive framework in most natural systems, especially in high-diversity systems such as tropical rainforests and coral reefs[48–50]. The quest for a general understanding of biological diversity would benefit from more experimental tests of which assembly regime—niche or dispersal—prevails in different contexts[48].

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

## Methods

### Study system

We conducted the study on granite rip-rap seawalls in Singapore. Singapore is a small (728.6 km²) tropical island city-state in Southeast Asia: it comprises a mainland (approximately 710 km²), which is approximately 50 km from east to west and 26 km from north to south, and 64 smaller offshore islands, including Sentosa island where the present study was conducted (specifically, at 1° 15′ 00.4″ N, 103° 49′ 04.0″ E). More than 80% of Singapore's natural coastline has been modified and replaced by granite rip-rap seawalls[51]. Natural rocky shores are now limited to a short 300-m stretch along the southern shore and a few islands south of Singapore's mainland (including parts of Sentosa island). Located just one degree above the equator, our study system experiences a typical tropical climate: high and uniform temperatures, abundant rainfall and high humidity all year round. For the duration of the study, from March 2021 to March 2022, the average reported temperature was 27.9 °C and the average monthly precipitation was 19.6 cm at our study site[52]. Facing the Singapore Strait, our study system is an open hard-bottomed system in a highly urbanized marine environment that experiences a mixed semi-diurnal tidal regime[51,53].

Monthly biodiversity surveys over a year conducted in ref. 36, which included our study location, found that, whereas composition of the communities colonizing seawalls and rocky shores differed, the two habitats harboured similar suites of species. This pattern is consistent with hard-bottom benthic systems around the world, including those in temperate and sub-tropical regions[24,54,55]. The study[36] also found no significant temporal patterns in species assemblages (that is, our study system can be considered aseasonal), with molluscs, crustaceans and algae dominating all year round (see also ref. 56). Although rocky shores often host greater species richness compared with seawalls, many of the species unique to either habitat are rare species: for instance, in their monthly surveys over a year, ref. 36 found less than ten individuals for all unique species in either habitat. What is notable is that, although seawalls are artificial, in the tropics they can still harbour relatively high species numbers: in Singapore, the authors of ref. 36 found a total of 138 species on natural rocky shores versus 105 species on seawalls through their year-long survey. Communities on aseasonal tropical hard shores also undergo relatively rapid rates of succession[27,57,58] compared with temperate and seasonal tropical hard shores[59,60].

The seawalls at our study location were constructed from boulders approximately 50 years ago and are not grouted with concrete or any other fill material[61]. These boulders are not regularly replaced or maintained, due to the ideal strength and material properties afforded by granite and the relatively low wave energy in the meso-tidal coastal environment of Singapore[51,62,63]. The biota on these hard structures, which act as the source pool for immigrants to our experimental units, can thus be considered climax communities[24,57].

### Habitat tile design and fabrication

Concrete tiles (Fig. 2a) measuring 0.2 m × 0.2 m × 0.06 m (width × length × depth) were used as habitat units from which we sampled our ecological communities. This approach is advantageous as it allows us to standardize habitat area and niche diversity across all replicates. Only one tile design was used and it comprised two microhabitat component types relevant to intertidal organisms: square pits and grooves[25,26,64]. These structural components varied in their sizes and depths from 0.01 m to 0.04 m; specifically, pits were first positioned randomly on the tile, after which we overlaid the remaining unoccupied space with grooves using a Truchet tiling algorithm. This was done using the three-dimensional modelling software Rhino 6. Once the tile design was finalized, the model was three-dimensional printed and from this rubber moulds were made. The actual experimental tiles were then cast from the moulds using concrete (1:3 cement to sand mix)[25]. During casting, four stainless-steel M8 hex nuts were embedded into the tile to act as insets for attaching the stainless-steel cage set-up to each tile unit using stainless-steel bolts (Fig. 2b). Temperature and light conditions in the experimental units were comparable with the surrounding seawall (Supplementary Fig. 1). Previous studies conducted worldwide[65] and in Singapore (including our study location; for example, refs. 25–27,57,58) have shown tiles to be an effective experimental device for sampling the benthic macrofaunal diversity of these tropical intertidal communities.

### Determining diameter of hole on panels

To determine the diameter of holes on panels used in the experimental set-up (Fig. 2), we conducted a field survey of the study site (before the start of the experiment, in February 2021), as well as an adjacent natural rocky shore site located further along the same shoreline approximately 1 km away. Five 1 m² quadrats were placed randomly along a 100 m transect at the mid-tidal height (approximately 0.5 m above chart datum; that is, along the same tidal height as in our experiment) at each site. From each quadrat, we counted the number of individuals of all macrofauna (greater than 1 mm) found within the 1 m² quadrat and measured the maximum body length of each individual using a digital Vernier caliper (with a resolution of 0.01 mm). A total of 245 individuals were recorded. The mean maximum body length of individuals was 16.8 mm (the mean was 15.0 mm for seawalls and 18.4 mm for rocky shores; Supplementary Fig. 2). Only two of the 245 individuals (less than 1%) had maximum body lengths greater than 40 mm (these two were gastropod species: *Chicoreus capucinus* (a whelk), measuring 42.8 mm, and *Peronia verruculata* (an onch slug), measuring 40.4 mm). Note, however, that their widths were less than 40 mm and that both of these species were present in the actual experimental communities.

### Data analyses

We fitted formulas arising from mechanistic mathematical models corresponding to each of the two hypotheses to the data on species richness $S$ versus number of holes $h$, using generalized nonlinear least squares (function gnls in the R programming language[66]). The classic model formula arises from a dynamic model describing how the rate of change of species richness on an island depends on the immigration and extinction rates, yielding the equilibrium solution[15,16,67].

$$S = \frac{\lambda P}{\lambda + \mu} \tag{1}$$

where $P$ is the number of species in the metacommunity, $\lambda$ is the per-species immigration rate and $\mu$ is the per-species extinction rate. In the first version of this model[15], the extinction rate is assumed to be inversely proportional to the area of the island ($\mu = \mu'/A$), giving

$$S = \frac{\lambda P}{\lambda + \frac{\mu'}{A}} \tag{2}$$

In our study system, area $A$ is constant and we assume that the immigration rate is proportional to the number of holes $h$, so that $\lambda = \alpha h$ for some parameter $\alpha$. Writing $\alpha' = \alpha A/\mu'$ then gives a two-parameter relationship between $S$ and $h$:

$$S = \frac{\alpha' h P}{\alpha' h + 1} \tag{3}$$

In the second version of this model[16,67], the extinction rate is, instead, assumed to be proportional to the number of species currently on the island ($\mu = \mu'' S$), giving

$$S = \frac{\lambda P}{\lambda + \mu'' S}$$

$$\Rightarrow S = \frac{-\lambda + \sqrt{\lambda\,(\lambda + 4P\mu'')}}{2\mu''} \tag{4}$$

Again setting $\lambda = \alpha h$ and writing $\alpha'' = \alpha/\mu''$, gives another two-parameter relationship between $S$ and $h$:

$$S = \frac{-\alpha'' h + \sqrt{\alpha'' h\,(\alpha'' h + 4P)}}{2} \tag{5}$$

Under the alternative new hypothesis, a mathematical formula for the relationship of species richness to number of holes arises from a master equation describing neutral drift within each of a fixed number of equal-sized non-overlapping niches in a local community[68,69]. Occasional stochastic extinction is balanced by immigration of new individuals from a larger metacommunity. The model has four parameters: the number of niches $n^*$, the local community size $J$, the probability that a new individual is an immigrant $m$ and the fundamental biodiversity number $\theta$ of the metacommunity from which immigrants are drawn. Species richness at the dynamic equilibrium is[3]

$$S \approx \theta\left\{ \psi_0\left(\frac{\theta}{n^*} + \gamma^*(\psi_0(\gamma^* + J^*) - \psi_0(\gamma^*))\right) - \psi_0\left(\frac{\theta}{n^*}\right) \right\} \tag{6}$$

where $\psi_0$ is the digamma function[70], $\gamma^* = (J^* - 1)m/(1 - m)$ is a composite immigration parameter and $J^* = J/n^*$ is the number of individuals in each niche. The original derivation of equation (6) assumed death–birth dynamics, where a death event leads to a vacated space that can be filled by the birth of a new individual, but the same equilibrium solution arises from birth–death dynamics where a newly born individual displaces and kills an established individual. Other variants of the standard model, for example, with abundance measured as biomass instead of number of individuals, lead to similar equilibrium solutions. To relate the composite immigration parameter $\gamma^*$ to the number of holes in our seawall experiment, we assume that the number of immigrants per unit time is proportional to the number of holes and that the number of local births in a given niche is proportional to the number of individuals residing there (minus one for the individual that just died to vacate a space), so that the probability of immigration is $m = bh/(bh + c(J^* - 1))$ for some constants $b$ and $c$, which leads to $\gamma^* = bh/(cn^*)$. We then define the composite parameter $\alpha = b/(cn^*)$ to get $\gamma^* = \alpha h$. Given data on $J$, the model then has three free parameters, $\alpha$, $n^*$ and $\theta$, which can be estimated from data on $S$ and $h$.

Note that in the new model the per-individual immigration rate scales linearly with number of holes, but in the classic model it is the per-species immigration rate that scales in this way. We argue that the novel model is more realistic because the per-species rate is likely to exhibit a convex relationship with number of holes (similar to a species accumulation curve). Modifying the classic model to allow a convex relationship would be likely to reduce the quality of its fit to the data (because it would be harder for it to capture the observed upswing in species richness at high numbers of holes), but we chose not to do this because the linear relationship is consistent with Wilson's original conceptualization[15] and because we wanted to conduct a conservative test of the new hypothesis.

We fitted each of the three models (two versions of the classic model, equations (3) and (5); and the new model, equation (6)) to the data by generalized least squares. After fitting each model, we performed 1,000 bootstraps stratified by hole treatments to obtain 95% confidence intervals on the parameter estimates and fitted values of species richness. The distribution of errors for each model was close to Gaussian in all cases (Supplementary Fig. 3).

## Reporting summary

Further information on research design is available in the Nature Portfolio Reporting Summary linked to this article.

## Data availability

The data generated in this study are available in the Zenodo repository: https://doi.org/10.5281/zenodo.7819940. Source data are provided with this paper.

## Code availability

The R code used for data analysis and for producing the figures are available in the Zenodo repository: https://doi.org/10.5281/zenodo.7819940.

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

**Acknowledgements** We thank E. Heery and N. Kristensen for helpful comments on the manuscript and H. Muller-Landau for helpful discussion. L.H.L.L. was supported by a Macquarie University Research Fellowship (no. 110042722) and a grant from Wildlife Reserves Singapore Conservation Fund. R.A.C. was supported by a grant from the James S. McDonnell Foundation (no. 220020470).

**Author contributions** L.H.L.L. conceived and conducted the experiment, collected the data, performed analyses and wrote the first draft of the manuscript. R.A.C. contributed ideas, performed analyses and edited the manuscript. All authors contributed substantially to revisions.

**Competing interests** The authors declare no competing interests.

**Additional information**
**Correspondence and requests for materials** should be addressed to Lynette H. L. Loke.

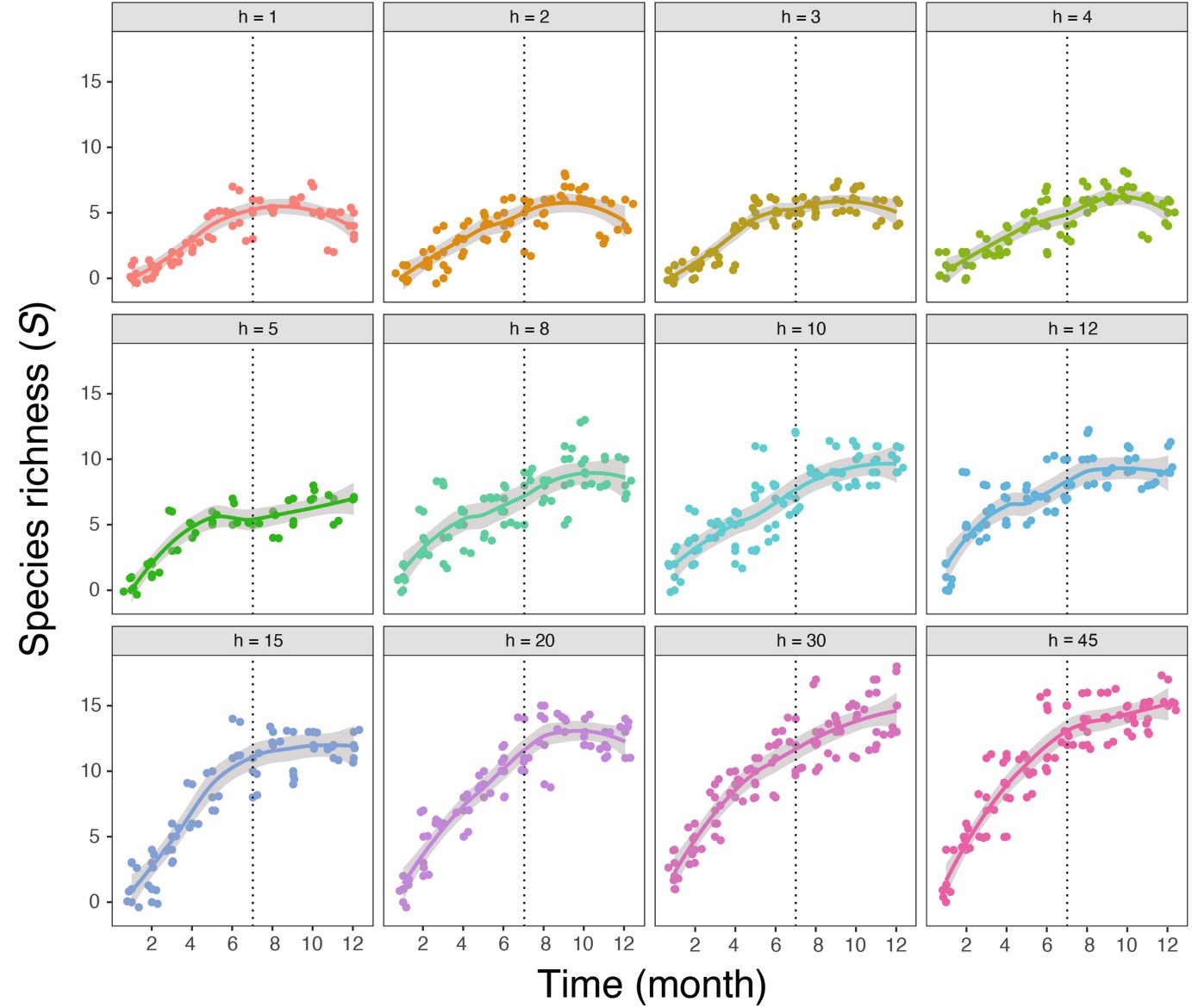

**Extended Data Fig. 1 | Trends in species richness in the experimental communities over a year.** Each plot shows species richness (*S*) against time (months) for one experimental treatment (panels), where treatments vary in the level of immigration (governed by number of holes, *h*). Experimental communities stabilized after 6–7 months for all but the highest immigration treatments. The horizontal and vertical coordinates of points have been jittered to improve visibility. Solid curves are mean estimates with shaded 95% confidence intervals; vertical dashed lines indicate the seventh month from the start of the experiment.

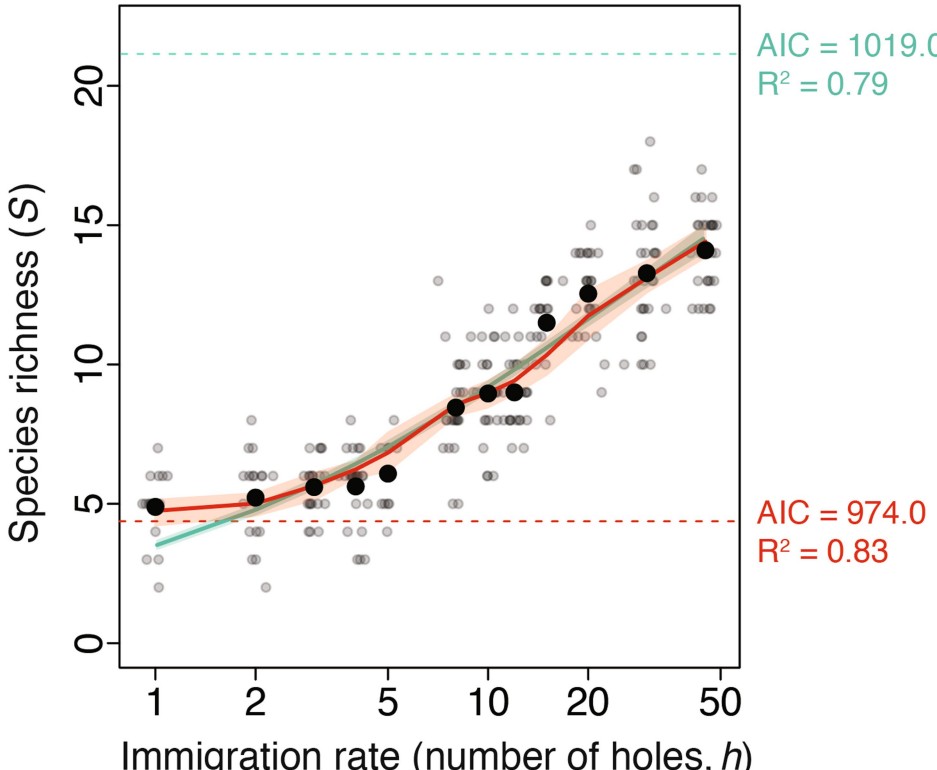

**Extended Data Fig. 2 | Fitting a variant of the classic model.** Each small grey point indicates a single experimental unit (seawall tile; 12 $h$-treatments, $n$ = 5) in a given month (x-coordinates are jittered to improve visibility); each large black point shows a treatment mean. Fitted models are shown by the solid curves (mean estimates) with shaded 95% confidence intervals from 1,000 bootstraps; the novel model (Eq. 6) is shown in red while the variant of the classic model is in green (Eq. 5; see Methods). This variant of the classic model was substantially poorer than the novel model although better than the original classic model (judged by AIC). The dashed green horizontal line indicates the expected species richness at high levels of immigration under the variant of the classic model, as estimated from the fitted parameters.

# Reporting Summary

## Statistics

For all statistical analyses, confirm that the following items are present in the figure legend, table legend, main text, or Methods section.

| n/a | Confirmed | |
|---|---|---|
| ☐ | ☒ | The exact sample size (*n*) for each experimental group/condition, given as a discrete number and unit of measurement |
| ☐ | ☒ | A statement on whether measurements were taken from distinct samples or whether the same sample was measured repeatedly |
| ☐ | ☒ | The statistical test(s) used AND whether they are one- or two-sided *Only common tests should be described solely by name; describe more complex techniques in the Methods section.* |
| ☒ | ☐ | A description of all covariates tested |
| ☐ | ☒ | A description of any assumptions or corrections, such as tests of normality and adjustment for multiple comparisons |
| ☐ | ☒ | A full description of the statistical parameters including central tendency (e.g. means) or other basic estimates (e.g. regression coefficient) AND variation (e.g. standard deviation) or associated estimates of uncertainty (e.g. confidence intervals) |
| ☒ | ☐ | For null hypothesis testing, the test statistic (e.g. *F*, *t*, *r*) with confidence intervals, effect sizes, degrees of freedom and *P* value noted *Give P values as exact values whenever suitable.* |
| ☒ | ☐ | For Bayesian analysis, information on the choice of priors and Markov chain Monte Carlo settings |
| ☒ | ☐ | For hierarchical and complex designs, identification of the appropriate level for tests and full reporting of outcomes |
| ☐ | ☒ | Estimates of effect sizes (e.g. Cohen's *d*, Pearson's *r*), indicating how they were calculated |

*Our web collection on statistics for biologists contains articles on many of the points above.*

## Software and code

Policy information about availability of computer code

Data collection
No software was used for data collection in this study.

Data analysis
All data analyses and figures were generated using R software version 3.5.1 as described in the Methods. Rhino software version 6 was used to design experimental tile in this study. All code used to analyse the data in this study are available in the the Zenodo repository at: 10.5281/zenodo.7819940 (see also 'Code availability' section).

For manuscripts utilizing custom algorithms or software that are central to the research but not yet described in published literature, software must be made available to editors and reviewers. We strongly encourage code deposition in a community repository (e.g. GitHub). See the Nature Portfolio guidelines for submitting code & software for further information.

## Data

Policy information about availability of data

All manuscripts must include a data availability statement. This statement should provide the following information, where applicable:
- Accession codes, unique identifiers, or web links for publicly available datasets
- A description of any restrictions on data availability
- For clinical datasets or third party data, please ensure that the statement adheres to our policy

The data underlying this study are available in the Zenodo repository at: 10.5281/zenodo.7819940

# Research involving human participants, their data, or biological material

Policy information about studies with human participants or human data. See also policy information about sex, gender (identity/presentation), and sexual orientation and race, ethnicity and racism.

| | |
|---|---|
| Reporting on sex and gender | n/a |
| Reporting on race, ethnicity, or other socially relevant groupings | n/a |
| Population characteristics | n/a |
| Recruitment | n/a |
| Ethics oversight | n/a |

Note that full information on the approval of the study protocol must also be provided in the manuscript.

# Field-specific reporting

Please select the one below that is the best fit for your research. If you are not sure, read the appropriate sections before making your selection.

☐ Life sciences ☐ Behavioural & social sciences ☒ Ecological, evolutionary & environmental sciences

For a reference copy of the document with all sections, see [nature.com/documents/nr-reporting-summary-flat.pdf](http://nature.com/documents/nr-reporting-summary-flat.pdf)

# Ecological, evolutionary & environmental sciences study design

All studies must disclose on these points even when the disclosure is negative.

| | |
|---|---|
| Study description | We conducted a manipulative field experiment involving the installation of 60 experimental setups or units (12 treatments × 5 replicates) on intertidal seawalls in Singapore (for more details please refer to the main text and Methods). |
| Research sample | The number of species, species identities, and abundances of the macrofaunal intertidal communities within each setup were censused every month for a year. Major taxa included gastropods, bivalves, polychaetes, tunicates and crustaceans (please refer to the Supplementary information for a full list of the benthic macroinvertebrate species recorded in this study). |
| Sampling strategy | Sample size selection was based on findings from previous studies done at the same study site and nearby locations, and determined by logistical feasibility (for more details please refer to the main text and Methods). |
| Data collection | Number of species, species identities, and abundances of the macrofaunal intertidal communities within each setup were censused by the lead and corresponding author every month for a year. |
| Timing and spatial scale | The study was conducted from from March 2021 to March 2022. Experimental setups were installed in randomised order along the base of intertidal seawalls across a 400 m stretch at our study site in Singapore, and maintained every 2–4 weeks during low tide. |
| Data exclusions | None. |
| Reproducibility | Specifications of our experimental setups and fabrication details are provided. Data and code to reproduce the analyses of the empirical data are available. |
| Randomization | All experimental setups were installed in randomised order along the base of intertidal seawalls at our study site. |
| Blinding | All setups were censused. |

Did the study involve field work? ☒ Yes ☐ No

# Field work, collection and transport

| | |
|---|---|
| Field conditions | Field work was conducted during low tide in clear weather conditions. For the duration of the study, the average reported temperature was 27.9°C and the average monthly precipitation was 19.6 cm at our study site. |
| Location | The study was conducted along the base of intertidal seawalls (~0.5 m above chart datum) specifically at 1°15'00.4"N, 103°49'04.0" E in Singapore. |

| | |
|---|---|
| Access & import/export | Research permits from the Singapore National Parks Board (NP/RP20-077) were obtained to conduct the research. No specimens were collected or removed from the field. We notified all relevant officer(s) at Sentosa Development Corporation before entering our study site to conduct the research. |
| Disturbance | Disturbance was minimal and all experimental setups were removed at the end of the experiment. |

# Reporting for specific materials, systems and methods

We require information from authors about some types of materials, experimental systems and methods used in many studies. Here, indicate whether each material, system or method listed is relevant to your study. If you are not sure if a list item applies to your research, read the appropriate section before selecting a response.

## Materials & experimental systems

| n/a | Involved in the study |
|---|---|
| ☒ | ☐ Antibodies |
| ☒ | ☐ Eukaryotic cell lines |
| ☒ | ☐ Palaeontology and archaeology |
| ☐ | ☒ Animals and other organisms |
| ☒ | ☐ Clinical data |
| ☒ | ☐ Dual use research of concern |
| ☒ | ☐ Plants |

## Methods

| n/a | Involved in the study |
|---|---|
| ☒ | ☐ ChIP-seq |
| ☒ | ☐ Flow cytometry |
| ☒ | ☐ MRI-based neuroimaging |

## Animals and other research organisms

Policy information about studies involving animals; ARRIVE guidelines recommended for reporting animal research, and Sex and Gender in Research

| | |
|---|---|
| Laboratory animals | The study did not involve laboratory animals. |
| Wild animals | Benthic macroinvertebrates (mainly gastropods, bivalves, polychaetes, tunicates and crustaceans; age unknown) were counted in the field and not captured or removed. |
| Reporting on sex | n/a |
| Field-collected samples | No specimens were collected or removed from the field. |
| Ethics oversight | None required. |

Note that full information on the approval of the study protocol must also be provided in the manuscript.

