## [Peer Review File · Nature]

Manuscript Title: Unveiling the transition from niche to dispersal assembly in ecology

Reviewer Comments & Author Rebuttals

Reviewer Reports on the Initial Version:

Referees' comments:

Referee #1 (Remarks to the Author):

The controversy between theories defending the prominent role of either dispersal and stochasticity or the stabilizing forces of species interactions in determining local species diversity has gone on and on for years. The debate goes back to the very founders of ecology as a field about a century ago. Since the question is extremely difficult to solve and answer in a general, simple way, most ecologists have no problem to agree that both dispersal and species interactions are important in driving local diversity, and that the relative role of each type of processes is, to a very large extent, fully scale- and/or context-dependent. The adventure to successfully analyze such an old, central question in ecology is reserved to very few people. This ms addresses this issue and claims that "most species are not stably coexisting but instead transiently co-occurring and reliant on continued immigration".

More precisely, the authors present an immigration/colonization experimental setup in intertidal communities that was designed to show that there is a clear transition from niche to dispersal assembly mechanisms at a particular immigration level. Over that immigration threshold, the community behaves as an assemblage of almost non-interacting species mostly sustained by external immigration. Only, under that threshold, at lower immigration rates, species diversity is stabilized by niche-driven species interactions.

The implications of these findings are huge. First, most ecological theory uses Lotka-Volterra-inspired models, which all rely on species interactions and niches. The variety of conclusions derived from this theoretical approach may be only valid for local communities in the low immigration regime. Second, our conservation efforts should be oriented to the preservation of the connectivity between local communities, since dispersal is the main mechanism responsible for maintaining species richness at the level we observe.

Although the authors clearly separate results from their experiment from speculative conclusions that would apply to ecological communities in general, I believe it is important to highlight the limitations of the current analysis of the conducted experiment in supporting more general conclusions.

My main concern has to do with the fact that the conclusions the authors derive stem from the analysis of a single pattern. This is clearly a weak point. The authors will be more convincing if they could show that the underlying theory is able to predict not only an immigration-species curve, as the one the authors carefully analyze, but also other typical community ecology patterns, such as

the similarity between replicates at the same immigration level and across immigration levels, the species abundance curve, or the very identity of species of the {em core} community, the one that is stabilized at the low-immigration regime. None of this is done and I wonder why. Capturing as many patterns as possible is what makes theory both beautiful and reliable.

The ms reads well. It is straightforward, and goes directly to show a single central result. The evidence of the transition from niche- to dispersal-assembly in the data analyzed is solid like a rock. I am not an expert on underwater marine experiments, so I cannot judge this part in detail. I don't have major concerns in relation to the appropriate use of statistics and treatment of uncertainties. I don't see much problems in relation to clarity and context in the abstract, introduction and conclusions section. Therefore, my comments will be minor and not too many. I hope the authors will find them useful.

`\section*{The Immigration-Species Curve}`

`\begin{itemize}`

`\item` The authors compare three curves: the one from their theory, which was derived in a previous publication, and two additional curves: the predicted curve by the standard model underlying MacArthur and Wilson Island Biogeography Theory (IBT), and a third curve that assumes that the extinction rate is proportional to the number of species currently on the island. This third curve does not rely on a third fundamentally different theory, but is only a slight modification of IBT based on a phenomenological assumption about how extinction rates depend on the species currently on the island. In this sense, the third one is only a phenomenological curve, while the first and the second are based on fundamentally different theories that are able to predict more community patterns than just the immigration-species curves. While phenomenological curves are designed to capture a single pattern, good theories are able to make predictions about more than one pattern because they are based on clear dynamic processes. This is why I think the paper will be strong enough and even more clear if only two curves (so two theories) are compared: the one suggested by the authors, which is able to capture the flattening of the curve at low immigration levels and the one directly derived from classic Island Biogeography Theory.

`\item` The authors present likelihood based AIC. This likelihood is based on a probability error, this is, the probability of observing certain deviation from the theory at a given immigration level. What are these errors look like? No much information is provided. Are they Gaussian?

`\item` I also miss a better description of the parameters and functions that appear in the curve derived by one of the authors in a previous publication. The digamma function is not even mentioned.

`\end{itemize}`

`\section*{Discussion}`

I only miss references on a body of literature that talks about the core-satellite hypothesis. According to this theory, when sites are sampled, all belonging to the same metacommunity, there are species that are always present in every site. This would be the core of the community. This core would be stabilized in every site through niche-based mechanisms. In addition, there is a subcommunity of satellite species only sustained by immigration. Most occupancy curves show that

in every local site, most species are satellite (see Hanski's book on Metapopulation Ecology). This resonates with authors' conclusion. It would be nice to make this connection.

David Alonso

Referee #2 (Remarks to the Author):

This is an intriguing paper. It is very well written and encompasses a simple yet ingenious experimental set up. The experiment was well conceived and performed (but see below). The authors conducted an experimental test to discriminate between two ecological niche theories using a manipulative field experiment on tropical seawalls. The authors proposed a new hypothesis which they found support for in their test - the relationship of species richness to immigration rate stabilised at a low value at low immigration rates but did not saturate at high immigration rates

To my knowledge the statistics are appropriate (however I am not an expert on such statistics). The conclusions are original and I am very confident that the results will be of general interest to Nature readers. It could make a very valuable contribution to the fields of both theoretical and applied ecology.

In saying all of that, my greatest criticism of this paper is the short-term nature of it (12 mo) and the knowledge that the authors are only observing the very early stages of succession, of animals only (undoubtedly algae and other non-animal taxa are also key components of this system? - fitting in with Eltonian niche theory and organisms interacting with one another and shaping their environment). In addition to that, surely the experimental units must act as inclusion cages for larger animals (despite having the potential to escape, perhaps this is harder for them) which may alter biotic interactions. Was there a reason that the authors did not include a control, whereby they just observed natural colonisation in the absence of the polycarbonate panels? Before considering this paper for publication, I would ask the authors to comment on the above points.

Referee #3 (Remarks to the Author):

The authors present the results of a clever and original experiment to test the effect of immigration rate on the diversity of benthic sea weeds and invertebrates in the intertidal zone near Singapore. Their hypothesis is that immigration will increase diversity above some critical threshold where niche space in the community becomes saturated. Immigration is manipulated by constructing boxes with varying numbers of holes to permit entry of settling recruits. Although I found the motivation for the experiment interesting and the approach clever, the overall story is not convincing enough to warrant publication in Nature, and the novelty of the idea is oversold. The paper should make a nice contribution to a regular ecology journal once the authors address these issues.

The first issue is with the novelty of the "novel" hypothesis. The novel hypothesis, described in refs 14+15, is a restatement of source-sink dynamics (as Pulliam called them in 1988) or "mass effects" (as Leibold et al. called them in 2004), and the idea has been applied to understand diversity since at

least Mouquet and Loreau (2003). The idea is that diversity can be super-saturated relative to potential stable coexistence based on niche space when the number of immigrants is high relative to local births and deaths. Calling this the “novel hypothesis” seems over-sold as the basic idea has been around in this form for at least 35 years. Fig. 1 shows the transitions between patch dynamics, species sorting and mass effects (to use Leibold et al.’s terminology) and is not really a novel take on the question of how immigration affects diversity.

Second, the authors argue that the number of holes in the exclosures affects immigration and not “Abiotic conditions” within the experimental units (L92-93). However, you need to read the Supplement to learn that the only conditions measured were temperature and light (via Hobo loggers), measured in only two of the treatments (low and medium immigration level) for only one month of the experiment. This seems like a minimal and inadequate test of effects of the treatments on abiotic conditions. From the illustration of the setup, it seems that the number of holes very likely affected water residence time over the experimental units, which likely affects biological oxygen demand, pH, and calcification, none of which were measured. It may be that the environment inside the low immigration treatments was stressful and this is why diversity was lower.

L113-114 is unclear, what else would immigration depend on?

Author Rebuttals to Initial Comments:

We thank the reviewers for their helpful comments. Our responses are in blue below (original reviewer comments are in black).

Referees' comments:

Referee #1 (Remarks to the Author):

The controversy between theories defending the prominent role of either *dispersal and stochasticity* or *the stabilizing forces of species interactions* in determining local species diversity has gone on and on for years. The debate goes back to the very founders of ecology as a field about a century ago. Since the question is extremely difficult to solve and answer in a general, simple way, most ecologists have no problem to agree that both dispersal and species interactions are important in driving local diversity, and that the relative role of each type of processes is, to a very large extent, fully scale- and/or context-dependent. The adventure to successfully analyze such an old, central question in ecology is reserved to very few people. This ms addresses this issue and claims that "most species are not stably coexisting but instead transiently co-occurring and reliant on continued immigration".

More precisely, the authors present an immigration/colonization experimental setup in intertidal communities that was designed to show that there is a clear transition from niche to dispersal assembly mechanisms at a particular immigration level. Over that immigration threshold, the community behaves as an assemblage of almost non-interacting species mostly sustained by external immigration. Only, under that threshold, at lower immigration rates, species diversity is stabilized by niche-driven species interactions.

The implications of these findings are huge. First, most ecological theory uses Lotka-Volterra-inspired models, which all rely on species interactions and niches. The variety of conclusions derived from this theoretical approach may be only valid for local communities in the low immigration regime. Second, our conservation efforts should be oriented to the preservation of the connectivity between local communities, since dispersal is the main mechanism responsible for maintaining species richness at the level we observe.

Although the authors clearly separate results from their experiment from speculative conclusions that would apply to ecological communities in general, I believe it is important to highlight the limitations of the current analysis of the conducted experiment in supporting more general conclusions.

We thank the reviewer for recognising the significance of the problem we investigated and the implications our findings. Note that in response to comments by Reviewer #3 we have now renamed our hypotheses to the "niche-ceiling hypothesis" (formerly the classic hypothesis) and the "niche-floor hypothesis" (formerly the novel hypothesis).

As the reviewer indicates, we clearly separated our intertidal experimental results from speculative conclusions applying to other ecological communities, and we agree it is in general

important to highlight such limitations. We have made further edits to this effect on Lines 236–238: “Efforts to protect biodiversity must consider large-scale landscape connectivity to be successful—although we encourage further experimental tests of the niche-floor hypothesis to establish the generality of this conclusion.”

We also suggest that the predictions of the niche-floor hypothesis need to be tested in other systems on Lines 179–182, 185–187, and 193–200.

My main concern has to do with the fact that the conclusions the authors derive stem from the analysis of a single pattern. This is clearly a weak point. The authors will be more convincing if they could show that the underlying theory is able to predict not only an immigration-species curve, as the one the authors carefully analyze, but also other typical community ecology patterns, such as the similarity between replicates at the same immigration level and across immigration levels, the species abundance curve, or the very identity of species of the *core* community, the one that is stabilized at the low-immigration regime. None of this is done and I wonder why. Capturing as many patterns as possible is what makes theory both beautiful and reliable.

Our aim was indeed to compare this one key pattern, i.e., the relationship between species richness and immigration rate, against the predictions of the niche-floor and niche-ceiling hypotheses. We chose this pattern because the two hypotheses make starkly contrasting predictions (Fig. 1c–d). No previous study has conducted such a test of these hypotheses against this pattern. Other biodiversity patterns are not subject to such starkly different predictions. Indeed, community ecologists have for decades been frustrated by the fact that different ecological theories make similar predictions for many biodiversity patterns (e.g., Chave et al. 2002). The species abundance distribution, which the reviewer mentions, in particular has little power for distinguishing between ecological theories. These hurdles to progress in community ecology motivated us to identify this key pattern that *is* subject to starkly contrasting predictions and to design an experiment specifically designed to exploit this (which involved carefully choosing the range of immigration treatments and number of replicates).

To illustrate our point, we performed a co-occurrence analysis (related to the reviewer’s point about the core community) using our dataset, and we found that the results were consistent with the niche-floor hypothesis (checkerboard scores were higher than the null model for the low-immigration treatments, indicating deviation from random assembly) but were not specific to it, i.e., we would not be able to rule out other hypotheses (e.g., pure dispersal-assembly theories with idiosyncratic metacommunity abundance distributions), unless perhaps we ran another experiment with a much larger sample size that was designed to test these other patterns. We explained this in the main text on Lines 204–211.

We emphasise that our experimental approach contrasts with the observational approach, which is widespread in community ecology. The value of such retrospective observational studies is that often a large amount of data is available to enable testing of multiple patterns,

along the lines of what the reviewer mentions (we are thinking, for example, of studies based on the ForestGEO plots). But limitations of observational studies include their reliance on retroactive prediction (rather than true prediction of unseen data) and the fact that the data may not be ideally suited to purpose (because they were collected before the study was conceived). Perhaps tellingly, after decades of such studies (e.g., focussed on the ForestGEO plots), there is still no real consensus about the conditions under which ecological systems are dispersal- versus niche-assembled.

What we have done goes back to the classic scientific method, where one first identifies a single key prediction that differs markedly across two theories, then designs a crucial experimental test and carries out the experiment. The obvious costs of this approach are that it requires a new experiment and substantial field or lab work, but the benefit is that because it is more targeted, it facilitates more decisive discrimination between the two theories. That said, we fully intend to pursue future experiments with this study system to conduct refined tests of biodiversity theory, including the patterns that the reviewer mentions. We have now added text to this effect on Lines 211–215: “For our study system, our estimate of the transition point between the two regimes ($h \approx 10$) can inform future the design of future studies focussed on one regime or the other. Our results can also inform sample size determination for future studies focussing on more-detailed biodiversity patterns including co-occurrence matrices and species abundance distributions.”

The ms reads well. It is straightforward, and goes directly to show a single central result. The evidence of the transition from niche- to dispersal-assembly in the data analyzed is solid like a rock. I am not an expert on underwater marine experiments, so I cannot judge this part in detail. I don't have major concerns in relation to the appropriate use of statistics and treatment of uncertainties. I don't see much problems in relation to clarity and context in the abstract, introduction and conclusions section. Therefore, my comments will be minor and not too many. I hope the authors will find them useful.

We thank the reviewer for the compliments.

The Immigration-Species Curve

- The authors compare three curves: the one from their theory, which was derived in a previous publication, and two additional curves: the predicted curve by the standard model underlying MacArthur and Wilson Island Biogeography Theory (IBT), and a third curve that assumes that the extinction rate is proportional to the number of species currently on the island. This third curve does not rely on a third fundamentally different theory, but is only a slight modification of IBT based on a phenomenological assumption about how extinction rates depend on the species currently on the island. In this sense, the third one is only a phenomenological curve, while the first and the second are based on fundamentally different theories that are able to predict more community patterns than just the immigration-species curves. While phenomenological curves are designed to capture a single pattern, good theories are able to make predictions about more than one pattern

because they are based on clear dynamic processes. This is why I think the paper will be strong enough and even more clear if only two curves (so two theories) are compared: the one suggested by the authors, which is able to capture the flattening of the curve at low immigration levels and the one directly derived from classic Island Biogeography Theory.

We thank the reviewer for the suggestion. We have now removed the third curve (i.e., a variant of the niche-ceiling hypothesis) in the main Figure (Fig. 3), and included it as an Extended data Figure instead (Extended data Fig. 2).

- The authors present likelihood based AIC. This likelihood is based on an probability error, this is, the probability of observing certain deviation from the theory at a given immigration level. What are these errors look like? No much information is provided. Are they Gaussian?

We thank the reviewer for the suggestion. We have now included a distribution of the errors for each model in a new Figure (Extended data Fig. 4). The errors were close to Gaussian in both cases (Lines 387–388).

Extended data Fig. 4. Distribution of the model errors. Errors (observed S – fitted S) for (a) the niche-ceiling model, and (b) the niche-floor model.

- I also miss a better description of the parameters and functions that appear in the curve derived by one of the authors in a previous publication. The digamma function is not even mentioned.

We agree that the technical details were not fully explained here. We have now explained (in the Methods) that ψ_0 is the digamma function, and have given a reference for it. We have also carefully explained how many parameters the model has (four) and what they all mean, how the composite parameters relate to the original parameters, and how we derived the relationship of the composite immigration parameter to the number of holes.

Excerpt from our revised Methods (Lines 352–371):

“The model has four parameters: the number of niches n^* , the local community size J , the probability that a new individual is an immigrant m , and the fundamental biodiversity number θ of the metacommunity from which immigrants are drawn. Species richness at the dynamic equilibrium is¹⁴

$$S \approx \theta \left\{ \psi_0 \left(\frac{\theta}{n^*} + \gamma^* (\psi_0(\gamma^* + J^*) - \psi_0(\gamma^*)) \right) - \psi_0 \left(\frac{\theta}{n^*} \right) \right\} \quad (6)$$

where ψ_0 is the digamma function⁷⁰, $\gamma^* = (J^* - 1)m/(1 - m)$ is a composite immigration parameter, and $J^* = J/n^*$ is the number of individuals in each niche. The original derivation of Eq. (6) assumed death–birth dynamics, where a death event leads to a vacated space that can be filled by the birth of a new individual, but the same equilibrium solution arises from birth–death dynamics where a newly born individual displaces and kills an established individual. Other variants of the standard model, e.g., with abundance measured as biomass instead of number of individuals, lead to similar equilibrium solutions. To relate the composite immigration parameter γ^* to the number of holes in our seawall experiment, we assume that the number of immigrants per unit time is proportional to the number of holes, and that the number of local births in a given niche is proportional to the number of individuals residing there (minus one for the individual that just died to vacate a space), so that the probability of immigration is $m = bh/(bh + c(J^* - 1))$ for some constants b and c , which leads to $\gamma^* = bh/(cn^*)$. We then define the composite parameter $\alpha = b/(cn^*)$ to get $\gamma^* = \alpha h$. Given data on J the model then has three free parameters: α , n^* and θ , which can be estimated from data on S and h .”

Discussion

I only miss references on a body of literature that talks about the core-satellite hypothesis. According to this theory, when sites are sampled, all belonging to the same metacommunity, there are species that are always present in every site. This would be the core of the community. This core would be stabilized in every site through niche-based mechanisms. In addition, there is a subcommunity of satellite species only sustained by immigration. Most occupancy curves show that in every local site, most species are satellite (see Hanski's book on Metapopulation Ecology). This resonates with authors' conclusion. It would be nice to make this connection.

We agree that the core–satellite hypothesis is related to our niche-floor hypothesis (previously the “novel hypothesis”). The details, however, depend on exactly how one interprets the core–

satellite hypothesis. On the one hand, in the mathematical models in classic papers on this topic (Hanski 1982; Hanski & Gyllenberg 1983), it is clear that, as immigration tends to zero, species richness in each local community also tends to zero, so there is no niche-assembled regime. In view of this, the core–satellite hypothesis would be better viewed as part of dispersal-assembly theory. Hanski (1982) appears to confirm this view when he writes (p216), “An alternative model... states that core species are better adapted to the environment than are satellite species”, implying that the core–satellite model does not involve local niche stabilisation, consistent with his mathematical models. On the other hand, later in the same paper he writes (p218) that, “if the core–satellite hypothesis is upheld, one may proceed by restricting the application of equilibrium theory to the core species, and employing appropriate non-equilibrium models for the satellite species.” Although the latter statement sounds very much like the niche-floor hypothesis, it is difficult to reconcile with his statement on p216, and Hanski (1982) does not actually build a mathematical model that incorporates attempts to unify equilibrium (i.e., niche-assembly) theory and non-equilibrium (i.e., dispersal-assembly) theory (and perhaps this is why his verbal statements on the subject are inconsistent and not fully rigorous). Moreover, he definitely does not propose the kind of biphasic species richness–immigration curve we show in Fig. 1d.

We conclude that although the core–satellite hypothesis contains ideas that are related to the niche-floor hypothesis, in classic papers on the core–satellite hypothesis the ideas are somewhat ambiguous and not fully formulated. We have not gone into detail on this topic in the manuscript, but we have added a mention of the core–satellite hypothesis on Line 55, along with mentions to other related work: “Related ideas go back to Diamond’s community assembly rules¹⁷, the core–satellite hypothesis¹⁸, and source–sink dynamics¹⁹.”

David Alonso

References

- Chave, J., Muller-Landau, H. C. & Levin, S. A. Comparing classical community models: theoretical consequences for patterns of diversity. *Am. Nat.* **159**, 1–23 (2002).
- Hanski, I. Dynamics of regional distribution: the core and satellite species hypothesis. *Oikos* **38**, 210–221 (1982).
- Hanski, I. & Gyllenberg, M. Two general metapopulation models and the core-satellite species hypothesis. *Am. Nat.* **142**, 17–41 (1983).

Referee #2 (Remarks to the Author):

This is an intriguing paper. It is very well written and encompasses a simple yet ingenious experimental set up. The experiment was well conceived and performed (but see below). The authors conducted an experimental test to discriminate between two ecological niche theories using a manipulative field experiment on tropical seawalls. The authors proposed a new hypothesis which they found support for in their test - the relationship of species richness to immigration rate stabilised at a low value at low immigration rates but did not saturate at high immigration rates

To my knowledge the statistics are appropriate (however I am not an expert on such statistics). The conclusions are original and I am very confident that the results will be of general interest to Nature readers. It could make a very valuable contribution to the fields of both theoretical and applied ecology.

We thank the reviewer for the compliments.

In saying all of that, my greatest criticism of this paper is the short-term nature of it (12 mo) and the knowledge that the authors are only observing the very early stages of succession, of animals only (undoubtedly algae and other non-animal taxa are also key components of this system? - fitting in with Eltonian niche theory and organisms interacting with one another and shaping their environment).

One advantage of our study system in the aseasonal tropical hard shores of Southeast Asia is that the rate of succession is very rapid, compared to most temperate hard shores. As we have shown in Extended data Fig. 1, species richness in our experimental communities stabilised after 6–7 months. Observed communities were also similar to the background seawall and no longer in the early stages of succession. This rapid succession has been documented in other studies of our system, on seawalls at the same tidal heights (i.e., mid-shore) and location (i.e., on Sentosa Island; see Hartanto et al. 2022), as well as at other nearby locations around Singapore (e.g., Loke et al. 2016; Loke & Todd 2016; Loke et al. 2017; Loke et al. 2019; Hsiung et al. 2019). In all these studies, early successional species were replaced within 1–2 months, and communities stabilised within 6 months. This is comparable to other studies of succession, conducted on the seasonal tropical rocky shores of Hong Kong; even on those shores, early stages of succession ended within 2–3 months irrespective of the presence or absence of herbivores, mucus treatments and location (Kaehler & Williams 1998; Williams et al. 2000) and recovery from major disturbances happened within 6 months (Hutchinson & Williams 2003). We have now made edits to further clarify this on Lines 279–281: “Communities on aseasonal tropical hard shores also undergo relatively rapid rates of succession^{26,57,58} compared to temperate and seasonal tropical hard shores^{59,60}.”

We agree with the reviewer that algal communities as well as other interacting biota are important components of the system. However, it was necessary to define the scope of our study to make it feasible. We therefore sought specifically to identify the number of intertidal

macrofaunal species that could coexist via niche mechanisms (i.e., without substantial immigration). Undoubtedly, if we included algae and other species, this number would increase, but we expect the main result of a niche-assembled to dispersal-assembled transition would be robust, i.e., the data would still qualitatively resemble Fig. 3. We also note that all biodiversity studies targeting questions of community assembly and coexistence restrict the scope of target taxa in some way (e.g., trees in the ForestGEO network, annual plants on serpentine soils or lizard communities on Caribbean islands, to give just a few well-known examples; see Davies et al. 2021; Levine & Hill 2009; Pringle et al. 2019). Furthermore, identifying algae to species level is currently not feasible in our system because of a lack of taxonomic work.

To make it clear that we are focussed on coexistence and community assembly of intertidal macrofaunal species specifically, we have added the following text on Lines 106–108: “Although we focused on macrofaunal intertidal species for practical reasons, we expect qualitatively similar results would be obtained if we included algal and microbial communities.”

In addition to that, surely the experimental units must act as inclusion cages for larger animals (despite having the potential to escape, perhaps this is harder for them) which may alter biotic interactions.

The size of the holes in our experimental units was 40 mm (note we mistakenly wrote 4 mm at one point in the original manuscript—we have corrected this now). In a survey we conducted at the study site prior to the start of experiment, only two of 245 organisms were larger than this, with the largest having body length 42.8 mm (see Extended data Fig. 3). Note, however, that their widths were less than 40 mm and both these species were present in the actual experimental communities (see Lines 321–323). This suggests that the vast majority of organisms in the study system could pass in and out of the holes. Principle grazers in our system were gastropods; larger species, such as echinoids typical of temperate rocky shores (Gaines & Lubchenco 1982) are absent from our system. During the entire duration of the experiment, we did not find any large-sized organisms trapped within the cages. Thus, we presume that any larger animal that is able to get into the cage would also have been able to get out. It is possible that the cages excluded larger organisms such as large fish (recall that the cages were submerged 95% of the time), but fish and other subtidal organisms were not part of our target community.

Was there a reason that the authors did not include a control, whereby they just observed natural colonisation in the absence of the polycarbonate panels? Before considering this paper for publication, I would ask the authors to comment on the above points.

We did not include a procedural control (i.e., cage without polycarbonate panels) because this was not a treatment–control study. Our experiment aimed to study how one variable (species richness) changes as a function of another variable (number of holes) instead. In other words, we wanted an experimental system where we could control immigration. Our experimental system is artificial. It was not our intention to try to compare our experimental communities

with the biodiversity of an unmodified system. Our goal was to see how species richness of our experimental system, which includes the cage, varies with immigration (as mediated by number of holes). A similar experiment could, for example, be conducted on lab-based microbial communities, in which case it is clear there is no natural analogue.

References

- Davies, S. J. et al. ForestGEO: Understanding forest diversity and dynamics through a global observatory network. *Biol. Conserv.* **253**, 108907 (2021).
- Gaines, S. D. & Lubchenco, J. A unified approach to marine plant-herbivore interactions. II. Biogeography. *Annu. Rev. Ecol. Evol. Syst.* **13**, 111–138 (1982).
- Hartanto, R. S. et al. (2022). Material type weakly affects algal colonisation but not macrofaunal community in an artificial intertidal habitat. *Ecol. Eng.* **176**, 106514.
- Hsiung, A. R. et al. Little evidence that lowering the pH of concrete supports greater biodiversity on tropical and temperate seawalls. *Mar. Ecol. Prog. Ser.* **656**, 193–205 (2020).
- Hutchinson, N. & Williams, G. A. Disturbance and subsequent recovery of mid-shore assemblages on seasonal, tropical, rocky shores. *Mar. Ecol. Prog. Ser.* **249**, 25–38 (2003).
- Kaehler, S. & Williams, G. A. Early development of algal assemblages under different regimes of physical and biotic factors on a seasonal tropical rocky shore. *Mar. Ecol. Prog. Ser.* **172**, 61–71 (1998).
- Levine, J. M. & HilleRisLambers, J. The importance of niches for the maintenance of species diversity. *Nature* **461**, 254–257 (2009).
- Loke, L. H. L., Bouma, T. J. & Todd, P. A. The effects of manipulating microhabitat size and variability on tropical seawall biodiversity: field and flume experiments. *J. Exp. Mar. Biol. Ecol.* **492**, 113–120 (2017).
- Loke, L. H. L., Chisholm, R. A. & Todd, P. A. Effects of habitat area and spatial configuration on biodiversity in an experimental intertidal community. *Ecology* **100**, e02757 (2019).
- Loke, L. H. L., Liao, L. M., Bouma, T. J. & Todd, P. A. Succession of seawall algal communities on artificial substrates. *Raffles Bull. Zool.* **32**, 1–10 (2016).
- Loke, L. H. L. & Todd, P. A. Structural complexity and component type increase intertidal biodiversity independently of area. *Ecology* **97**, 383–393 (2016).

Pringle, R. M. et al. Predator-induced collapse of niche structure and species coexistence. *Nature* **570**, 58–64 (2019).

Williams, G. A., Davies, M. S. & Nagarkar, S. Primary succession on a seasonal tropical rocky shore: the relative roles of spatial heterogeneity and herbivory. *Mar. Ecol. Prog. Ser.* **203**, 81–94 (2000).

Referee #3 (Remarks to the Author):

The authors present the results of a clever and original experiment to test the effect of immigration rate on the diversity of benthic sea weeds and invertebrates in the intertidal zone near Singapore. Their hypothesis is that immigration will increase diversity above some critical threshold where niche space in the community becomes saturated. Immigration is manipulated by constructing boxes with varying numbers of holes to permit entry of settling recruits. Although I found the motivation for the experiment interesting and the approach clever, the overall story is not convincing enough to warrant publication in *Nature*, and the novelty of the idea is oversold. The paper should make a nice contribution to a regular ecology journal once the authors address these issues.

The first issue is with the novelty of the “novel” hypothesis. The novel hypothesis, described in refs 14+15, is a restatement of source-sink dynamics (as Pulliam called them in 1988) or “mass effects” (as Leibold et al. called them in 2004), and the idea has been applied to understand diversity since at least Mouquet and Loreau (2003). The idea is that diversity can be super-saturated relative to potential stable coexistence based on niche space when the number of immigrants is high relative to local births and deaths. Calling this the “novel hypothesis” seems over-sold as the basic idea has been around in this form for at least 35 years. Fig. 1 shows the transitions between patch dynamics, species sorting and mass effects (to use Leibold et al.’s terminology) and is not really a novel take on the question of how immigration affects diversity.

We thank the reviewer for recognising the merits of our experiment. The reviewer seems to be agreeing with our interpretation of the results but is concerned that our “novel” hypothesis is not sufficiently novel. To this we have two responses. Firstly, the main novelty of our study was the experiment, not the hypothesis. Secondly, and most importantly, there was another hypothesis (our “classic” hypothesis) that made distinctly different predictions, and it was not obvious a priori which hypothesis was correct—this was the motivation for the experiment. Although both of the hypotheses have long intellectual heritages in ecology (as the reviewer notes for the “novel” hypothesis), and although each make starkly different predictions about the relationship of species richness to immigration, no-one had conducted this crucial experimental test until now.

To address the reviewer’s concern, we have renamed our “novel hypothesis” to “niche-floor hypothesis”; we have also renamed our “classic hypothesis” to “niche-ceiling hypothesis”. Having made this change, we do nevertheless emphasise that no study prior to Chisholm & Fung (2021) had brought all the pieces together to produce the theoretical species richness versus immigration curve shown in Fig. 1d, which is the key pattern tested by our experiment (against the alternative niche-ceiling hypothesis in Fig. 1c). We agree that the source-sink hypothesis could lead to a curve similar to the one shown in Fig. 1d, although we are unaware of any theoretical study that has actually generated such a prediction and certainly of no experimental study that might have tested it. We have added a mention of the source-sink hypothesis near the beginning of the manuscript on Lines 55–56: “Related ideas go back to Diamond’s community assembly rules¹⁷, the core-satellite hypothesis¹⁸, and source-sink

dynamics¹⁹.” We have added another mention near the end on Lines 189–193: “By contrast, the dispersal-assembly explanation attributes high local tree species richness to immigration, as well as potentially larger-scale niche processes (e.g., source–sink dynamics¹⁹, with species being adapted to distinct habitats across the landscape), and ultimately to diversification processes that play out over geological timescales and continental spatial scales.”

We agree with the reviewer’s comment that “Fig. 1 shows the transitions between patch dynamics, species sorting and mass effects”, in that the former two are examples of niche-assembly and the latter is an example of dispersal-assembly (Shmida & Wilson 1985). In particular, we agree that Fig. 1d can be interpreted as a transition from patch-dynamics or species-sorting to mass effects. But we also argue that these paradigms are just examples of the more general paradigms of niche-assembly (the former two) and dispersal-assembly (the latter). We do, however, recognise that “mass effects” will be very familiar to many ecologists, so we have added a reference to mass effects in the first paragraph of our main text on Lines 32–35 where we discuss dispersal-assembly theory: “Under dispersal-assembly theory, including MacArthur & Wilson’s theory of island biogeography^{1,2}, the neutral theory of biodiversity³, and mass effects⁴, local diversity arises from an immigration–extinction balance, and is ultimately dependent on regional processes.” We have also mentioned mass effects in the Discussion on Lines 181–182: “implying that most species are not stably coexisting but instead transiently co-occurring and reliant on continued immigration⁵ (i.e., mass effects⁴).” We have not mentioned patch dynamics and species sorting, simply because there is an extremely large body of literature on niche-assembly theory and we cannot mention every potentially relevant concept.

Second, the authors argue that the number of holes in the enclosures affects immigration and not “Abiotic conditions” within the experimental units (L92-93). However, you need to read the Supplement to learn that the only conditions measured were temperature and light (via Hobo loggers), measured in only two of the treatments (low and medium immigration level) for only one month of the experiment. This seems like a minimal and inadequate test of effects of the treatments on abiotic conditions. From the illustration of the setup, it seems that the number of holes very likely affected water residence time over the experimental units, which likely affects biological oxygen demand, pH, and calcification, none of which were measured. It may be that the environment inside the low immigration treatments was stressful and this is why diversity was lower.

We would first like to clarify that the size of the holes in our experimental units was 40 mm; we mistakenly wrote 4 mm at one point in the original manuscript but we have corrected this now. We also emphasise that our setups were submerged at high tide 95% of the time, and so we expected light and temperature conditions to be relatively homogeneous most of the time. Nevertheless, we continued to measure light and temperature because compared to other abiotic parameters, these are the two with the greatest potential to influence the experimental communities (Raffaelli & Hawkins 1996; Little et al. 2009). We chose to measure these two parameters only in the low and median immigration treatments because, if there were any significant differences between conditions inside and outside the experimental units, they

would presumably be stronger in the low-to-median range of treatments, where the number of holes was lower. But, as expected, we found minimal differences in temperature and light levels between the inside and outside of the low to median setups (Appendix S1). We measured these conditions over a month because the trends in deviations were close to zero after this time and our study location was an aseasonal environment with uniform temperatures all year round (Line 260).

We did not measure pH (or relatedly calcium content) because we established in a previous study (i.e., Hsiung et al. 2019) conducted at several locations, including one that was approximately 1 km from our present study location along the same shoreline, that even large differences in pH have no measurable effect on the experimental seawall communities. Specifically in that study we used a technique called concrete carbonation to achieve a close-to-neutral pH of 7–8 and compared that with concrete tiles with an alkaline pH of 9–13. That study also ran for a year and we found no significant difference in the species richness, abundance or composition of the colonising assemblage on the concrete tiles in all locations (in Singapore and the UK). Differences in pH also had no effect on barnacle or limpet cover and algal succession from 6 to 12 months (Hsiung et al. 2019; also the duration of our present study).

We did not measure biological oxygen demand (BOD) or dissolved oxygen as our system was an open hard-bottom system in a high wave-energy environment which experiences a mixed semi-diurnal tidal regime (Van Maren & Gerritsen 2012). Even in the low-immigration treatments, water is completely drained at low tide during sampling, implying that water in the experimental units completely turns over at least twice a day. In a previous study (i.e., Loke et al. 2017), also conducted in Singapore, we found that small-scale hydrodynamic differences caused by the topography of concrete tiles had no measurable effect on the colonising assemblage.

Furthermore, we conjecture that if conditions were truly more stressful in the low-immigration treatments, any resulting systematic error in species richness would tend to work against our hypothesis. Specifically, stressful conditions would lower reduce species richness in the low-immigration treatments (relative to the trend extrapolated from the high-immigration treatments) and make the richness versus immigration curve look more like Fig. 2c rather than Fig. 2d. However, we found that species richness plateaued for low numbers of holes (Fig. 3), consistent with Fig. 2d. If the issues raised by the reviewer do exist, they must be minimal.

Nevertheless, we acknowledge that these same questions about abiotic conditions inside the experimental units may occur to other readers and we thank the reviewer for drawing this to our attention. We have reworded the text on Lines 96–97: “Temperature and light conditions in the experimental units were comparable to the surrounding seawall (Supplementary Fig. 1).”, and added the following text to the Methods on Lines 263–265: “Facing the Singapore Strait, our study system is an open hard-bottomed system in a highly urbanised marine environment which experiences a mixed semi-diurnal tidal regime^{51,53}.” We also added the following text to the Supplementary Information (Lines 742–745): “We did not measure pH or calcium content as it has been established in a previous study⁵⁸, conducted at several locations including one

that was approximately 1 km from our present study location along the same shoreline, that even large differences in pH have no measurable effect on the experimental seawall communities we studied.”

L113-114 is unclear, what else would immigration depend on?

We meant to emphasise that our assumption was that immigration was proportional, i.e., linearly related, to the number of holes, not just that immigration was dependent on the number of holes. We have now replaced “proportional” with “linearly related” to make this more explicit (Lines 118–121): “We assumed that immigration was linearly related to the number of holes (h), a parsimonious assumption that, if relaxed, only strengthens our results (see Methods).”

References

Chisholm, R. A. & Fung, T. Examining the generality of the biphasic transition from niche-structured to immigration-structured communities. *Theor. Ecol.* **15**, 1–16 (2022).

Hsiung, A. R. et al. Little evidence that lowering the pH of concrete supports greater biodiversity on tropical and temperate seawalls. *Mar. Ecol. Prog. Ser.* **656**, 193–205 (2020).

Little, C., Williams, G. A. & Trowbridge, C. D. *The Biology of Rocky Shores 2nd Edition* (Oxford University Press, 2009).

Loke, L. H. L., Bouma, T. J. & Todd, P. A. The effects of manipulating microhabitat size and variability on tropical seawall biodiversity: field and flume experiments. *J. Exp. Mar. Biol. Ecol.* **492**, 113–120 (2017).

Raffaelli, D. & Hawkins, S. J. *Intertidal Ecology* (Chapman & Hall, 1996).

Shmida, A. V. I. & Wilson, M. V. Biological determinants of species diversity. *J. Biogeogr.* **12**, 1–20 (1985).

Van Maren, D. S. & Gerritsen, H. Residual flow and tidal asymmetry in the Singapore Strait, with implications for resuspension and residual transport of sediment. *J. Geophys. Res.* **117**, C04021 (2012).

Reviewer Reports on the First Revision:

Referees' comments:

Referee #1 (Remarks to the Author):

Dear Authors,

I have read the whole rebuttal letter in detail and the paper again. I found very interesting authors responses and the reading of the comments and concerns by the other reviewers.

In ecology, very often, several verbal hypotheses are used to account for similar patterns. In my opinion, their coexistence and the variety and idiosyncrasy of their different names make a weak favor to the field. For instance, why do we have "mass effects" and "dispersal-assembly" as two different hypotheses if they mean and involve very similar processes? Why? Let me try an answer. The first reason is because they are only/mainly verbal. If both of them were well anchored in a mathematical formulation, we could either tease them apart or end up deciding they are exactly the same. In the latter case, as it is done with synonym species names, we ecologists should keep only one official name for the same thing. Incidentally, in my opinion, for this particular example, dispersal-assembly is a better/older name for this particular hypothesis. The second reason is a little bit sociological. You may not agree with it, but I think there is something true in it. It is easy for a good and creative writer, probably native-English speaker, and well-established ecologist to come up with a seemingly new verbal hypothesis with a catchy name that correspond to an old hypothesis by verbally describing the same thing with little different nuances in a slight different context and no mathematical formulation at all.

Why do I make this introduction? Well, I don't see a strong reason for the authors to invent a new name for the hypothesis they are testing. They are testing a transition between an immigration regime that is consistent with niche effects stabilizing a core low-diversity community and an immigration regime that is consistent with dispersal-assembly. I don't like "niche-ceiling" vs "niche-floor hypothesis". Why do authors introduce this funny name? Referee 1 and 2 did not have any problem with the initial formulation. It seems to me it is only to make referee 3 happy that the authors decided to invent this name. This is not a strong reason. It makes again a weak favor to the field. What the authors need to discuss, as they already did, is their experiment in the light of previous contrasting hypothesis about how ecological communities get assembled. This is enough. This is simple. This is better. I encourage the authors to do it in spite of the third referee, who seems to think anyway that is a great paper, but not a Nature paper but only good enough for a specific journal in the area of ecology.

With regard to the authors response to my comments on the Hanski's Core-Satellite hypothesis, I think the authors are right when they say Hanski did not formulate rigorously and mathematically his hypothesis, but I have the feeling he would agree with the use of this terminology to describe a transition between these two immigration regimes the authors account for with their experiment. What else could stabilize the same low-diversity core community across sites at low immigration regimes if there were no biologically or environmentally well-defined niches?

In sum, according to a variety of metrics, many worse and better papers than the one we are handling now have been published in the ecology section of the Nature journal over the years. Of course, we will find Nature readers, as the third referee, that will think this is the kind of paper that could be published in any specific ecological journal. However, this is true for any Nature paper too.

In sum, I encourage the authors to think carefully if they really need to introduce new terminology and rewrite their discussion accordingly. They should not miss the opportunity to criticize previous terminology and hypotheses and make recommendations about keeping their number low, and restricting the addition of new terminology only when is very well justified. I believe that the experimental setting, the quantitative representation of their hypothesis, the connection between theory predictions and data, and the clear transition between two immigration regimes they finally report deserve attention and it is not usually found in typical ecology papers.

Referee #2 (Remarks to the Author):

I am satisfied that the authors have dealt with all of my comments sufficiently. I recommend this paper for publication.

Referee #3 (Remarks to the Author):

This revision and response to comments did not change my assessment of this paper- it is a clever experiment but did not adequately show that the dispersal treatment only affected dispersal, and the novelty is oversold. First, the test of differences in light and temperature are between the treatments with 5 and 15 holes, but the number of holes ranges from 1-45. Your test of the effect of the treatment on abiotic environment covers <25% of the range of the treatment, and only two variables. The main result of the paper is that the treatments with 1 & 2 holes have more species than you would expect from a linear relationship between holes and species (Fig. 3). It may be, therefore, that the water residence time at low tides is longer in these treatments in ways that affect chemistry, temperature, etc (and therefore richness, your main result). You would never know that from your data. You don't know that holes did not affect the environment in the treatments that drive your main result.

I also don't agree that higher stress in the lowest immigration treatment would "work against our hypothesis. Specifically, stressful conditions would lower reduce species richness in the low-immigration treatments". That assumes that higher stress results in lower richness, which any ecologist will tell you need not be the case if, for instance, stress reduces competition with dominant species.

Finally, I agree that the experimental design is novel and clever, but still did not feel that I was reading something fundamentally new. First, I don't agree that "no study prior to Chisholm & Fung (2021) had brought all the pieces together to produce the theoretical species richness versus immigration curve shown in Fig. 1d". Look at Fig. 3 in Mouquet and Loreau (2003), it shows local and beta diversity vs. immigration with transitions between niche and source sink dynamics. Second, here is a list of experimental studies showing immigration effects on local richness that I knew about without digging into the literature.

Kunin WE. 1998. Biodiversity at the Edge: A Test of the Importance of Spatial 'Mass Effects' in the Rothamsted Park Grass Experiments. *Proceedings of the National Academy of Sciences of the United States of America*. 95, pp. 207-212

Furey, George N., Hawthorne, Peter L., and Tilman, David. 2022. " Might Field Experiments Also Be Inadvertent Metacommunities?." *Ecology* 103(7): e3694. <https://doi.org/10.1002/ecy.3694>

Wandrag EM, Dunham AE, Duncan RP, Rogers HS. Seed dispersal increases local species richness and reduces spatial turnover of tropical tree seedlings. *Proc Natl Acad Sci U S A*. 2017 Oct 3;114(40):10689-10694. doi: 10.1073/pnas.1709584114.

Germain RM, Strauss SY, Gilbert B. Experimental dispersal reveals characteristic scales of biodiversity in a natural landscape. *Proc Natl Acad Sci U S A*. 2017 Apr 25;114(17):4447-4452. doi: 10.1073/pnas.1615338114.

Author Rebuttals to First Revision:

Referees' comments:

Referee #1 (Remarks to the Author):

Dear Authors,

I have read the whole rebuttal letter in detail and the paper again. I found very interesting authors responses and the reading of the comments and concerns by the other reviewers.

In ecology, very often, several verbal hypotheses are used to account for similar patterns. In my opinion, their coexistence and the variety and idiosyncrasy of their different names make a weak favor to the field. For instance, why do we have "mass effects" and "dispersal-assembly" as two different hypotheses if they mean and involve very similar processes? Why? Let me try an answer. The first reason is because they are only/mainly verbal. If both of them were well anchored in a mathematical formulation, we could either tease them apart or end up deciding they are exactly the same. In the latter case, as it is done with synonym species names, we ecologists should keep only one official name for the same thing. Incidentally, in my opinion, for this particular example, dispersal-assembly is a better/older name for this particular hypothesis. The second reason is a little bit sociological. You may not agree with it, but I think there is something true in it. It is easy for a good and creative writer, probably native-English speaker, and well-established ecologist to come up with a seemingly new verbal hypothesis with a catchy name that correspond to an old hypothesis by verbally describing the same thing with little different nuances in a slight different context and no mathematical formulation at all.

Why do I make this introduction? Well, I don't see a strong reason for the authors to invent a new name for the hypothesis they are testing. They are testing a transition between an immigration regime that is consistent with niche effects stabilizing a core low-diversity community and an immigration regime that is consistent with dispersal-assembly. I don't like "niche-ceiling" vs "niche-floor hypothesis". Why do authors introduce this funny name? Referee 1 and 2 did not have any problem with the initial formulation. It seems to me it is only to make referee 3 happy that the authors decided to invent this name. This is not a strong reason. It makes again a weak favor to the field. What the authors need to discuss, as they already did, is their experiment in the light of previous contrasting hypothesis about how ecological communities get assembled. This is enough. This is simple. This is better. I encourage the authors to do it in spite of the third referee, who seems to think anyway that is a great paper, but not a Nature paper but only good enough for a specific journal in the area of ecology.

We thank the referee for the thoughtful comments on this matter. We did indeed come up with the terminology of "niche-ceiling" and "niche-floor" to address Referee #3's concerns. Although we like the new terminology and find it quite evocative, we are very sympathetic to the referee's point about the proliferation of redundant terminology in the ecological literature, and we agree that the imperative not to further pollute the ecological namespace should take precedence here. Therefore, we have now reverted to our original description of the hypotheses, that is, "classic" vs. "novel".

With regard to the authors response to my comments on the Hanski's Core-Satellite hypothesis, I think the authors are right when they say Hanski did not formulate rigorously and mathematically his hypothesis, but I have the feeling he would agree with the use of this terminology to describe a transition between these two immigration regimes the authors account for with their experiment. What else could stabilize the same low-diversity core community across sites at low immigration regimes if there were no biologically or environmentally well-defined niches?

We thank the referee for acknowledging our previous response and agree that this is the most parsimonious interpretation of the core–satellite hypothesis which we still reference on Line 57.

In sum, according to a variety of metrics, many worse and better papers than the one we are handling now have been published in the ecology section of the Nature journal over the years. Of course, we will find Nature readers, as the third referee, that will think this is the kind of paper that could be published in any specific ecological journal. However, this is true for any Nature paper too.

In sum, I encourage the authors to think carefully if they really need to introduce new terminology and rewrite their discussion accordingly. They should not missed the opportunity to criticize previous terminology and hypotheses and make recommendations about keeping their number low, and restricting the addition of new terminology only when is very well justified. I believe that the experimental setting, the quantitative representation of their hypothesis, the connection between theory predictions and data, and the clear transition between two immigration regimes they finally report deserve attention and it is not usually found in typical ecology papers.

We thank the referee for the compliments, and for recognising the significance of our work. As mentioned previously, we agree with the view that ecology is plagued with an overabundance of terminology and have now reverted our description of the hypotheses to our original version, that is, “classic” vs. “novel”.

Referee #2 (Remarks to the Author):

I am satisfied that the authors have dealt with all of my comments sufficiently. I recommend this paper for publication.

We thank the referee for their time and constructive input.

Referee #3 (Remarks to the Author):

This revision and response to comments did not change my assessment of this paper- it is a clever experiment but did not adequately show that the dispersal treatment only affected dispersal, and the novelty is oversold.

First, the test of differences in light and temperature are between the treatments with 5 and 15 holes, but the number of holes ranges from 1-45. Your test of the effect of the treatment on abiotic environment covers <25% of the range of the treatment, and only two variables. The main result of the paper is that the treatments with 1 & 2 holes have more species than you would expect from a linear relationship between holes and species (Fig. 3). It may be, therefore, that the water residence time at low tides is longer in these treatments in ways that affect chemistry, temperature, etc (and therefore richness, your main result). You would never know that from your data. You don't know that holes did not affect the environment in the treatments that drive your main result.

We agree that it is conceivable that abiotic effects (e.g., differences in light, temperature and chemistry) could have played an additional role in influencing our results but we expect this influence to be minimal, and more importantly, not a result of longer "water residence time at low tides" in low-h treatments. Water residence time does not significantly differ among treatments at low tide because water is completely drained at low tide. As the tide recedes, water level within the setups across all treatments also equilibrates very quickly (i.e., within seconds) relative to the outside. Further, as we highlighted previously, our setups were exposed at low tide only 5% of the time. Therefore, it is not feasible that longer water residence time in low-h treatments could have resulted in markedly different abiotic conditions and consequently species richness. Although it is possible in principle that abiotic variables (i.e., temperature, light, and chemistry) could have differed among treatments during the short period when the setups were exposed at low tide (5% of the time), we expect the effects of to be minimal based on multiple lines of evidence, as previously explained:

- The temperature and light differentials between the inside and outside of the units was similar in the low- and median-immigration setups (Supplementary Fig. 1); trends in deviations were close to zero after a month and our study location was an aseasonal environment with uniform temperatures all year round.
- Previous studies we have conducted at various (including nearby) locations, have shown that even large differences in pH have no measurable effect on the species richness of our experimental seawall communities (Hsiung et al. 2019). So we can rule out the potential influence of pH differences among treatments on our results.
- In a previous study (i.e., Loke et al. 2017), also conducted in Singapore, we found that small-scale hydrodynamic differences caused by topography of the complex concrete tiles have no measurable effect on the species richness of our experimental seawall communities.
- A meta-analysis of 46 studies around the world (i.e., Dodds et al. 2022) as well as one study conducted in Singapore at the same study location (i.e., Hartanto et al. 2022) have also shown that material type has no measurable effect on the species richness of experimental seawall communities.

Nevertheless, we acknowledge that it is conceivable that abiotic differences could have played an additional minor role in influencing our results, and have now added the following caveat in the main text and Supplementary Information:

Discussion, Lines 217–220: “One assumption of our method is consistency of abiotic conditions across experimental immigration treatments. We confirmed this for light and temperature (see Methods), but it remains conceivable that other variables (e.g., pH) are important, and we encourage future similar experiments to measure as wide a range of abiotic variables as possible.”

Supplementary Information, page 4: “We did not measure the potential influence of other variable such as pH, calcium content or material type as it has been established in a previous studies^{1–3}, conducted at several locations worldwide, including one that was approximately 1 km from our present study location along the same shoreline, that differences in pH and material type have no measurable effect on the species richness of the experimental seawall communities we studied.

“We also conjecture that if there were substantial variation in unmeasured abiotic conditions, the tendency would be for these conditions to be more stressful in the low-immigration treatments which would tend to lower species richness in these treatments (relative to the trend extrapolated from the high-immigration treatments). Any resulting systematic error would thus tend to work against the novel hypothesis, by making the richness versus immigration curve look more like Fig. 1c than Fig. 1d.”

We also made the following edits:

“Efforts to protect biodiversity must account for large-scale landscape connectivity to be successful—although we encourage further experimental tests of the novel hypothesis in other systems and locations to establish the generality of this conclusion.”

References

Dodds, K. C. et al. Material type influences the abundance but not richness of colonising organisms on marine structures. *J. Environ. Manage.* **307**, 114549 (2022).

Hartanto, R. S. et al. Material type weakly affects algal colonisation but not macrofaunal community in an artificial intertidal habitat. *Ecol. Eng.* **176**, 106514 (2022).

Hsiung, A. R. et al. Little evidence that lowering the pH of concrete supports greater biodiversity on tropical and temperate seawalls. *Mar. Ecol. Prog. Ser.* **656**, 193–205 (2020).

Loke, L. H. L., Bouma, T. J. & Todd, P. A. The effects of manipulating microhabitat size and variability on tropical seawall biodiversity: field and flume experiments. *J. Exp. Mar. Biol. Ecol.* **492**, 113–120 (2017).

I also don't agree that higher stress in the lowest immigration treatment would "work against our hypothesis. Specifically, stressful conditions would lower reduce species richness in the low-immigration treatments". That assumes that higher stress results in lower richness, which any ecologist will tell you need not be the case if, for instance, stress reduces competition with dominant species.

The referee is concerned about the potential impact of more-stressful abiotic conditions in the low-immigration treatments. However, as noted above, multiple lines of evidence suggest that there was no substantial variation in abiotic conditions between treatments. Also, we note that the referee's comments on this matter are to some degree inconsistent: in the initial review, the referee wrote that "It may be that the environment inside the low immigration treatments was stressful and this is why diversity was lower"; but in the comment above, the referee writes that stress could increase diversity. Regardless, we agree with the reviewer that in principle there could be some minimal effect of variation in abiotic conditions between treatments on the results. We have now acknowledged this and added the following caveat in the main text (Lines 217–220) and Supplementary Information (see quotes above).

Finally, I agree that the experimental design is novel and clever, but still did not feel that I was reading something fundamentally new. First, I don't agree that "no study prior to Chisholm & Fung (2021) had brought all the pieces together to produce the theoretical species richness versus immigration curve shown in Fig. 1d".

Look at Fig. 3 in Mouquet and Loreau (2003), it shows local and beta diversity vs. immigration with transitions between niche and source sink dynamics. Second, here is a list of experimental studies showing immigration effects on local richness that I knew about without digging into the literature.

Kunin WE. 1998. Biodiversity at the Edge: A Test of the Importance of Spatial 'Mass Effects' in the Rothamsted Park Grass Experiments. *Proceedings of the National Academy of Sciences of the United States of America*. 95, pp. 207-212

Furey, George N., Hawthorne, Peter L., and Tilman, David. 2022. " Might Field Experiments Also Be Inadvertent Metacommunities?." *Ecology* 103(7): e3694. <https://doi.org/10.1002/ecy.3694>

Wandrag EM, Dunham AE, Duncan RP, Rogers HS. Seed dispersal increases local species richness and reduces spatial turnover of tropical tree seedlings. *Proc Natl Acad Sci U S A*. 2017 Oct 3;114(40):10689-10694. doi: 10.1073/pnas.1709584114.

Germain RM, Strauss SY, Gilbert B. Experimental dispersal reveals characteristic scales of biodiversity in a natural landscape. *Proc Natl Acad Sci U S A*. 2017 Apr 25;114(17):4447-4452. doi: 10.1073/pnas.1615338114.

Referee #3 agrees that our experimental design is “novel and clever” but claims that our idea is not “something fundamentally new”, and then refers to several past studies exploring the effect of immigration on local community species richness. We agree that the *general* idea of exploring the effect of immigration on local species richness is not novel: the novelty of our manuscript is to focus on the specific prediction of a biphasic relationship of species richness to immigration with a flat niche-structured phase at low immigration rates and an increasing immigration-structured phase at high immigration rates, and to contrast this with the classic prediction of a saturating relationship. After reading all the papers cited by the referee, we stand by our statement that “no study prior to Chisholm & Fung (2021) had brought all the pieces together to produce the theoretical species richness versus immigration curve”. To better articulate the novelty of the study, we discuss each of these papers in more detail below and have added the following sentence in the main text:

Lines 80–83: “Although ecologists acknowledge that immigration can influence community species richness^{2,7,11}, our experiment is the first to systematically vary immigration over a wide range in this way to reveal the functional form of the species richness versus immigration relationship and thus elucidate underlying mechanisms.”

Mouquet & Loreau (2003, Am. Nat.)

The referee is correct that this theoretical paper explored the connection between species richness and dispersal/immigration, as we have. However, the mechanisms in the model are different from ours, and the resulting relationship between richness and dispersal is also very different. The referee directs us to Figure 3 in the paper, which shows that local species richness (alpha diversity) in the model exhibits a hump-shaped relationship to immigration, as opposed to the biphasic relationship in our model and data. The reason Mouquet & Loreau get a hump-shaped relationship is as follows:

- They do not have stabilising niches in their model, and thus there is no flat niche-structured phase at low immigration, as we see in our model and our data.
- They have fixed fitness differences between species (that also vary across sites), that leads to a collapse in diversity when immigration is very high, which we do not see in our model or our data.

In summary, although the theoretical paper by Mouquet & Loreau does explore the same general topic as us (effects of immigration on species richness), it uses a different model and makes different predictions, a crucial reason being that it omits any mechanism for stabilising niches, which is essential in our model.

Kunin (1998, PNAS) and Furey et al. (2022, Ecology)

These two studies are both based on classic grassland experiments that were originally set up to test the role of niches (generally defined) on plant species richness. Both studies retrospectively explore whether immigration between neighbouring plots may have contaminated the experimental results, and find that indeed it has. The Kunin study looks at the

classic Rothamsted grassland experiment, which was set up to test the effects of fertilisation on plant growth, and finds that species richness is slightly higher at subplot edges when neighbouring plots are more dissimilar, suggesting an influence of dispersal (mass effects). The Furey et al. study focuses on the Cedar Creek grassland experiment and again finds that plots whose neighbours have more distinct species tend to have higher species richness than would otherwise be expected.

Although there is certainly general conceptual overlap with our study here, these studies are addressing a much narrower question, i.e., whether a plot's richness is boosted by higher richness in neighbouring plots. Their papers do not even touch on the topic of the overall relationship of species richness to immigration rate, let alone predict or test for a biphasic relationship. Their experimental setups were originally established without immigration in mind (perhaps symptomatic of the general historical neglect of the importance of immigration in ecology), but these studies themselves are retrospective and thus observational: immigration was not systematically varied across subplots. These are examples of studies that consider some aspect of the combined effects of immigration and niches on local ecological communities, but the similarity with our study ends there. If anything, these studies motivate the need for an experimental study like ours that systematically varies immigration.

Wandrag et al. (2017, PNAS)

This study looks at seedling diversity in forests of Guam, where there is currently little seed dispersal (due to extirpation of birds by introduced brown tree snakes), and in forests of nearby Saipan and Rota islands, where there is substantial seed dispersal (the snakes have not been introduced there). They find higher seedling diversity in Saipan and Rota, suggesting an important role for dispersal in increasing forest stand diversity. Because this is an observational study, however, there was no possibility of varying immigration systematically to determine the functional relationship of diversity to immigration (as in the grassland studies discussed above). It is precisely because of results like this that an experiment such as ours was needed.

Germain et al. (2017, PNAS)

This study is the most similar to ours in that they experimentally manipulate immigration into local communities, in this case serpentine grassland plots. They find a positive relationship of species richness to immigration, as we do, but crucially they do not uncover the niche-structured phase at low immigration. The reason for this is that their immigration treatments were based on augmentation, and they did not have a means for reducing immigration below natural levels, which, based on our theoretical expectations and our experiment, is necessary for revealing the niche-structured phase. Thus, this study is interesting but motivates the need for a study such as ours that experimentally manipulates immigration to arbitrarily low levels, in our case using our innovative exclusion devices on seawalls.

Another difference between the Germain et al. study and ours is that they manipulate immigration in terms of the diversity of propagules while keeping numbers of propagules

roughly constant, while we manipulate number of propagules. Both approaches can be informative about the effects of immigration on diversity in different ecological contexts.

On a technical note, we observe that Germain et al. claim that they observe no effect of immigration on species richness at the smallest and largest scales, which superficially might seem to have some connection with our results. But this is an artefact of them treating scale as a categorical variable in their Fig. 1. Their scale categories are 1 m, 5 m, 100 m, 5 km, 10 km. It is clear that the quantitative differences in scale are relatively small at the two extreme ends of this range. If the results from Fig. 1A are re-plotted with scale as a numerical variable, the relationship is clearly a continuously increasing one (see figure below; the dashed line shows a fit of y versus $\log(x)$, giving $R^2=0.992$ and $AIC=11.5$; the solid curve shows a fitted power law with vertical offset to account for baseline species richness, giving $R^2=0.997$ and $AIC=8.3$; both of these monotonically increasing relationships provide excellent fits, with the power-law fit being somewhat better). Thus Germain et al. do not uncover the niche-structured phase.